# ScaleDiff: Scaling Difficult Problems for Advanced Mathematical Reasoning

## Abstract

Large Reasoning Models (LRMs) have shown impressive capabilities in complex problem-solving, often benefiting from training on difficult mathematical problems that stimulate intricate reasoning. Recent efforts have explored automated synthesis of mathematical problems by prompting proprietary models or large-scale open-source models from seed data or inherent mathematical concepts. However, scaling up these methods remains challenging due to their high computational/API cost, complexity of prompting, and limited difficulty level of the generated problems. To overcome these limitations, we propose ScaleDiff, a simple yet effective pipeline designed to scale the creation of difficult problems. We efficiently identify difficult problems from existing datasets with only a single forward pass using an adaptive thinking model, which can perceive problem difficulty and automatically switch between "Thinking" and "NoThinking" modes. We then train a specialized difficult problem generator (DiffGen-8B) on this filtered difficult data, which can produce new difficult problems in large scale, eliminating the need for complex, per-instance prompting and its associated high API costs. Fine-tuning Qwen2.5-Math-7B-Instruct on the ScaleDiff-Math dataset yields a substantial performance increase of 11.3% compared to the original dataset and achieves a 65.9% average accuracy on AIME'24, AIME'25, HMMT-Feb'25, BRUMO'25, and MATH500, outperforming recent strong LRMs like OpenThinker3. Notably, this performance is achieved using the cost-efficient Qwen3-8B model as a teacher, demonstrating that our pipeline can effectively transfer advanced reasoning capabilities without relying on larger, more expensive teacher models. We also observe a clear scaling phenomenon in model performance on difficult benchmarks as the quantity of difficult problems increases. Our code is available at the anonymous repository https://anonymous.4open.science/r/ScaleDiff-D053.

## 1 Introduction

Recent advancements in Large Reasoning Models (LRMs) such as OpenAI-o1 (OpenAI, 2024) and DeepSeek-R1 (Guo et al., 2025) have demonstrated remarkable progress in tackling complex reasoning problems. These models exhibit the ability to perform trial-and-error, self-reflection, and iterative refinement within long Chains of Thought (CoT), leading to enhanced problem-solving capabilities. To replicate this success, various efforts have been made, employing techniques like Supervised Fine-Tuning (SFT) on distilled data (Tian et al., 2025; Moshkov et al., 2025; Team, 2025; Guha et al., 2025), Reinforcement Learning (RL) with verifiable rewards (Yu et al., 2025b; Zeng et al., 2025; Luo et al., 2025; He et al., 2025b), or more complex training pipelines based on SFT and RL (Face, 2025; Wen et al., 2025b; Chen et al., 2025; Liu et al., 2025b; Huang et al., 2025a). A common strategy among these approaches is to identify challenging mathematical problems from existing datasets for training (Moshkov et al., 2025; Wen et al., 2025b; Liu et al., 2025b). The rationale behind this is that difficult problems typically necessitate intricate reasoning processes, thereby stimulating more sophisticated model behaviors, whereas simpler problems often yield limited benefits. However, creating such difficult mathematical problems—particularly those at the competition or olympiad level—is often costly because they are primarily handcrafted by human experts (AI-MO; Lin, 2025; He et al., 2024). Recent research has explored the automated synthesis of mathematical data by prompting proprietary models as well as large-scale open-source counterparts, either from

seed data (Luo et al., 2023; Yu et al., 2024; 2025a; Toshniwal et al.) or from inherent mathematical concepts (Huang et al., 2025b; Tang et al., 2024; Zhao et al., 2025; Zhan et al., 2025). However, scaling these approaches remains challenging due to their substantial computational costs, complex prompting design, and relatively limited difficulty of the generated problems.

To further investigate the impact of difficult problems on enhancing complex reasoning abilities of LRMs, we propose ScaleDiff, a simple yet effective pipeline that scales the creation of difficult problems to improve models' complex reasoning capabilities. We begin by leveraging an existing adaptive thinking model (Zhang et al., 2025a), which can automatically switch between the "Thinking" and "NoThinking" modes depending on the difficulty of a given problem, thereby serving as a difficult problem identifier to detect difficult problems within existing datasets. This identification process requires only a single forward pass, making it more efficient than commonly used approaches such as *fail rate* and *LLM-as-a-judge*. Subsequently, to enable the generation of an arbitrary number of difficult problems, we train a problem generator (denoted as DiffGen-8B) on these identified difficult problems. We then utilize DiffGen-8B to generate large-scale new difficult problems, eliminating the need for complicated prompting design, per-instance shot selection, and the substantial computational costs required by traditional methods. For each generated problem, we distill its long CoT solution using Qwen3-8B (Yang et al., 2025) in "Thinking" mode. This comparatively small model provides solutions in a cost-efficient manner and offers a favorable alternative to widely used larger models such as DeepSeek-R1 or QwQ-32B. We also apply both rule and model filtration to these solutions. The final ScaleDiff-Math dataset is composed of these difficult problem-solution pairs and the original dataset.

Further SFT of Qwen2.5-Math-7B-Instruct on the ScaleDiff-Math dataset demonstrates promising performance. Our ScaleDiff consistently outperforms recent strong LRMs such as OpenThinker3 (Guha et al., 2025) and AceReason-Nemotron (Chen et al., 2025) across AIME'24 (AIMO), AIME'25 (Lin, 2025), HMMT Feb'25 (HMMT, 2025), BRUMO (BRUMO, 2025), and MATH500 (Lightman et al., 2023) on average. ScaleDiff also improves upon AM-Qwen3-Distilled-7B by enhancing both the difficulty and scale of the training data, resulting in a relative performance gain of 11.3%. These results highlighting the effectiveness of our approach in strengthing models' complex reasoning abilities. Moreover, by varying the size of the augmenting dataset, we observe a clear scaling phenomenon in model performance on AIME'24 and AIME'25, with accuracy improving as the number of difficult problems increased. This scaling behavior further highlights the potential of our method to drive continued gains as larger and more challenging datasets become available.

## 2 RELATED WORK

### 2.1 MATHEMATICAL DATA FOR LRMS

Numerous datasets have been proposed to enhance the mathematical reasoning capabilities of LRMs through SFT. Prevalent strategies (He et al., 2025b; Li et al., 2024b; Amini et al., 2019) involve *selecting data from existing sources* such as textbooks, examinations and websites. Beyond simple selection, some research focuses on data augmentation of these existing resources. *Answer augmentation* methods (Moshkov et al., 2025; Toshniwal et al.; Tong et al., 2024; Pan et al., 2025a; Toshniwal et al., 2024; Lin et al., 2025; Wang et al., 2025) use a teacher model to synthesize novel and diverse solutions for existing problems, aiming to boost the student model's performance. These methods are often referred to as data distillation. In contrast, *Question augmentation* methods (Luo et al., 2023; Yu et al., 2024; Toshniwal et al.; Li et al., 2024a; Lu et al., 2024; Mitra et al., 2024; Pei et al., 2025; Pan et al., 2025b) involve synthesizing novel problems and their corresponding solutions. This method can expand topical coverage, introduce more diverse problem structures, though it requires rigorous validation to ensure the correctness of synthesized questions and solutions (Yu et al., 2025a; Li et al., 2024c). To further enhance the diversity of synthetic data, *Persona-based augmentation* technique (Ge et al., 2024; Lambert et al., 2024; Li et al., 2023; Luo et al., 2024) has emerged. By incorporating role-playing into prompts, LLMs can generate diverse, role-specific mathematical problems. However, while some efforts have emerged to synthesize new questions, they do not explicitly control the data difficulty. Consequently, the generated problems often lack sufficient challenge for current top-tier LRMs, leading to limited improvements.

## 2.2 DIFFICULTY-AWARE DATA SELECTION AND SYNTHESIS

Data difficulty is a crucial metric for assessing data quality, significantly impacting the training effectiveness (Chen et al.). Previous research has explored *difficulty-aware question selection*. For example, S1 (Muennighoff et al., 2025) and Light-R1 (Wen et al., 2025a) filter out simple problems that small models can easily solve, retaining difficult ones for SFT. AceReason (Chen et al., 2025) further incorporates difficulty-based filtering into RL training. DeepMath-103K (He et al., 2025b) proposes a new dataset with a higher proportion of challenging problems. However, these methods are limited to selecting from existing data and cannot generate new, challenging examples. Furthermore, most of these techniques assess difficulty by *fail rate*, a model-specific metric, which may restrict their generalizability across different models.

Another line of research focuses on *synthesizing new data with varying difficulty*. In the mathematical domain, DART-Math (Tong et al., 2024) synthesizes more solutions for difficult problems, enhancing response diversity for challenging questions. MATH$^2$ (Shah et al., 2024) extracts core "skills" from existing math datasets and employs them as the basis for generating novel and difficult questions by prompting LLMs. DAST (Xue et al., 2025) proposes a difficulty-matching few-shot prompting method, presenting longer, more detailed examples for harder questions. ScaleQuest (Ding et al., 2024a) introduces Question Preference Optimization (QPO), which optimizes generated mathematical problems based on solvability and difficulty. The optimized questions then serve as positive samples for preference optimization of the question generator. MathSmith (Zhan et al., 2025) generates math problems from scratch using concept–explanation pairs, achieving superior performance on olympiad-level benchmarks. However, most of these methods are not specifically designed for synthesizing difficult math problems. The difficulty of their generated questions remains limited, leading to restricted performance improvements.

## 3 METHOD

In this section, we first introduce our identification of difficult problems in Section 3.1. We then detail our approach for generating a large-scale set of new challenging problems in Section 3.2. Finally, in Section 3.3, we describe our process for distilling and filtering high-quality solutions to these generated problems. The overview of our ScaleDiff pipeline is shown in Figure 1.

### 3.1 DIFFICULT PROBLEM IDENTIFICATION

Assessing the difficulty of mathematical problems primarily relies on two existing methods: *fail rate* (Tong et al., 2024) and *LLM-as-a-judge* (Gao et al.). Specifically, for *fail rate*, a proxy mathematical model is used to solve a given problem multiple times, and the proportion of incorrect responses determines its fail rate. For *LLM-as-a-judge*, a more powerful LLM is prompted with the mathematical problem, its reference solution (if available), and predefined criteria for difficulty assessment. However, both methods have their limitations: the *fail rate* is computationally inefficient as it necessitates multiple solution attempts by the proxy mathematical model; *LLM-as-a-judge* is highly sensitive to the specific rules and criteria predefined within the input prompt.

Different from existing methods, we seek to leverage *AdaptThink*[1] (Zhang et al., 2025a) as our difficult problem identifier. *AdaptThink* algorithm is designed to teach models to adaptively choose between a time-consuming "Thinking" process for complex problems and a direct "NoThinking" response for simpler ones through RL. This adaptive mechanism inherently reflects the model's perceived difficulty of a problem. The primary objective of *AdaptThink* is a constrained optimization:

$$\max \mathbb{E}_{(x,\cdot)\sim\mathcal{D},\, y\sim\pi_\theta(\cdot|x)}\mathbb{I}(y_1 = \texttt{</think>})),$$
$$\text{s.t. } \mathbb{E}_{(x,\cdot)\sim\mathcal{D},\, y\sim\pi_\theta(\cdot|x)}R(x,y) \geq \mathbb{E}_{(x,\cdot)\sim\mathcal{D},\, y'\sim\pi_{\theta_{ref}}(\cdot|x)}R(x,y'). \tag{1}$$

where $\mathcal{D}$ denotes the problem-solution dataset, and we let $\mathcal{P} = \{x \mid (x,y) \in \mathcal{D}\}$ denote its problem set. $\mathbb{I}(y_1 = \texttt{</think>})$ is an indicator function for the first generated token being $\texttt{</think>}$. $R(x,y)$ is the reward function representing the accuracy of the model's response for problem $x$ (returning 1 for a correct solution and 0 for an incorrect one). This objective aims to maximize the probability of generating "NoThinking" responses, subject to a constraint: the current model's

---

[1] https://huggingface.co/THU-KEG/AdaptThink-7B-delta0.05

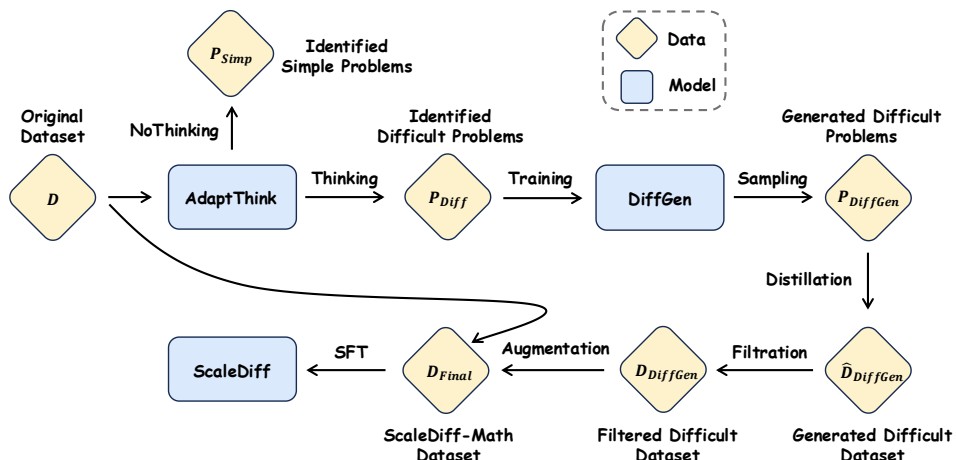

Figure 1: Overview of ScaleDiff pipeline. AdaptThink first identify difficult problems by deciding whether the model should invoke explicit reasoning. The difficult subset is then used to train DiffGen-8B, which generates additional challenging problems. Their solutions are distilled from a teacher model and filtered for quality, before being combined with the original data to augment SFT.

expected accuracy—the reward for a correct solution $R(x, y)$—must be maintained at or above that of a fixed reference model. This design compels *AdaptThink* to opt for the efficient "NoThinking" mode only when it does not compromise accuracy. Conversely, for problems where "NoThinking" would lead to a significant performance drop, *AdaptThink* is driven to engage its "Thinking" mode to satisfy the accuracy constraint.

This adaptive behavior effectively transforms *AdaptThink* into a binary classifier for problem difficulty. We define a problem $x$ as "simple" if *AdaptThink* produces a "NoThinking" response, and "difficult" otherwise. Formally, our problem identification criteria are based on the first generated token ($y_1$) of *AdaptThink*'s response:

$$\text{Difficulty}(x) = \begin{cases} \text{Simple} & \text{if } y_1 = \texttt{</think>} \\ \text{Difficult} & \text{if } y_1 \neq \texttt{</think>} \end{cases} \tag{2}$$

Notably, determining whether a problem is simple or difficult requires only the model's output of a first token (one forward pass), making the identification process highly efficient than *fail rate* and *LLM-as-a-judge*. By applying *AdaptThink* as such a identifier, we efficiently identify and extract challenging problem-solution pairs from existing datasets, forming a curated subset denoted as $\mathcal{D}_{\text{Diff}}$.

### 3.1.1 EFFECTIVENESS OF DIFFICULTY VIA PASS@K UNDER MULTIPLE MODELS

To validate that the difficulty split reflects genuine differences in problem-solving complexity, we evaluate Pass@k performance on problems from $\mathcal{D}_{\text{Simp}}$ and $\mathcal{D}_{\text{Diff}}$ under multiple models. Specifically, we randomly sample 2K problems from each subset and evaluate three models with short and long CoT settings: Qwen2.5-Math-7B-Instruct, Qwen3-8B in both "NoThinking" and "Thinking" modes, and DeepSeek-R1-Distill-Qwen-7B. We set $k = 3$ for all experiments. As shown in Table 1,

| Dataset | CoT Type | $\mathcal{D}_{\text{Simp}}$-2K | $\mathcal{D}_{\text{Diff}}$-2K |
|---|---|---|---|
| Qwen2.5-Math-7B-Instruct | Short | 88.2 | 65.8 |
| Qwen3-8B (NoThinking) | Short | 85.8 | 71.5 |
| Qwen3-8B (Thinking) | Long | 91.1 | 86.0 |
| DeepSeek-R1-Distill-Qwen-7B | Long | 93.4 | 85.0 |

Table 1: Pass@3 performance comparison between the $\mathcal{D}_{\text{Simp}}$-2K and $\mathcal{D}_{\text{Diff}}$-2K subsets across multiple models and CoT types.

two clear conclusions can be drawn: (1) All models exhibit substantially lower Pass@3 on the $\mathcal{D}_{\text{Diff}}$-2K subset compared with the $\mathcal{D}_{\text{Simp}}$-2K subset. (2) The performance gap is especially pronounced for short-CoT models, which tend to be more sensitive to reasoning difficulty.

### 3.1.2 Effectiveness of Difficulty via SFT Performance

To evaluate the validity of using *AdaptThink* as a difficult problem identifier and the effectiveness of difficult problems as training data, we conduct SFT on Qwen2.5-Math-7B-Instruct (Yang et al., 2024c) with the full dataset $\mathcal{D}$, as well as its two subsets: the identified difficult subset $\mathcal{D}_{\text{Diff}}$ and simple subset $\mathcal{D}_{\text{Simp}}$. To further control for data size, we downsample $\mathcal{D}_{\text{Simp}}$ to match the size of $\mathcal{D}_{\text{Diff}}$ and additionally construct a random subset $\mathcal{D}_{\text{Rand}}$ by sampling 192K problems from $\mathcal{D}$. To further examine the effect of diversity, we construct a mixed subset $\mathcal{D}_{\text{Mix}}$, which consists of all samples in $\mathcal{D}_{\text{Simp}}$ together with 20% of the samples from $\mathcal{D}_{\text{Diff}}$. More experimental details are provided in Section 4.1. The corresponding data size and results on three mathematical benchmarks are presented in Table 2. We also list the size of difficult sample and $\text{Div}_{\text{Global}}$ for each subset, with more details introduced in Appendix Section C.

| Model | Size | Difficult Size | Div$_{\text{Global}}$ | AIME'24 avg@10 | AIME'25 avg@10 | HMMT-Feb'25 avg@10 | BRUMO'25 avg@10 | MATH500 avg@3 | AVG |
|---|---|---|---|---|---|---|---|---|---|
| $\mathcal{D}$ | 558K | 192K | 0.497 | 63.0$_{\pm3.5}$ | 51.7$_{\pm5.6}$ | 33.3$_{\pm5.8}$ | 60.7$_{\pm7.7}$ | 94.6$_{\pm0.4}$ | 59.2 |
| $\mathcal{D}_{\text{Simp}}$ | 366K | 0 | 0.495 | 43.7$_{\pm5.3}$ | 38.7$_{\pm4.5}$ | 27.0$_{\pm5.3}$ | 53.7$_{\pm6.0}$ | 91.3$_{\pm0.4}$ | 48.9 |
| $\mathcal{D}_{\text{Simp}}$ | 192K | 0 | 0.476 | 40.7$_{\pm4.9}$ | 33.7$_{\pm2.8}$ | 24.0$_{\pm3.6}$ | 48.3$_{\pm7.8}$ | 90.4$_{\pm0.7}$ | 45.1 |
| $\mathcal{D}_{\text{Mix}}$ | 404K | 38K | 0.497 | 53.7$_{\pm4.3}$ | 41.3$_{\pm7.0}$ | 30.0$_{\pm4.7}$ | 55.3$_{\pm6.2}$ | 93.5$_{\pm0.3}$ | 52.5 |
| $\mathcal{D}_{\text{Rand}}$ | 192K | 66K | 0.497 | 54.3$_{\pm5.0}$ | 42.0$_{\pm5.2}$ | 31.3$_{\pm4.8}$ | 57.0$_{\pm4.6}$ | 93.2$_{\pm0.4}$ | 53.3 |
| $\mathcal{D}_{\text{Diff}}$ | 192K | 192K | 0.473 | 62.3$_{\pm5.0}$ | 44.3$_{\pm7.6}$ | 36.0$_{\pm5.7}$ | 59.0$_{\pm6.3}$ | 93.9$_{\pm1.2}$ | 56.6 |

Table 2: Effect of problem difficulty on SFT performance across three mathematical benchmarks.

It can be observed that training on the difficult subset $\mathcal{D}_{\text{Diff}}$ (192K) outperforms training on the simple subset $\mathcal{D}_{\text{Simp}}$ (56.6 vs. 45.1 on average) and randomly sampled subset $\mathcal{D}_{\text{Rand}}$ (56.6 vs. 53.3) of the same size , highlighting that difficult problems provide more effective training signals for enhancing reasoning ability. Even when comparing $\mathcal{D}_{\text{Diff}}$ (192K) against the much larger $\mathcal{D}_{\text{Simp}}$ (366K), the difficult subset still yields a clear advantage (56.6 vs. 48.9). Moreover, while training on the full dataset $\mathcal{D}$ (558K) achieves the strongest results overall (59.2), this improvement stems primarily from its larger scale. Notably, the performance gap between $\mathcal{D}_{\text{Diff}}$ (192K) and the full dataset is only 2.6 points (56.6 vs. 59.2), despite the latter being nearly three times larger. In contrast, the gap between $\mathcal{D}_{\text{Simp}}$ (192K) and the full dataset is approximately 14 points (45.1 vs. 59.2), underscoring that simple problems contribute far less effectively to improving model performance compared to difficult ones.

We further validate the effectiveness of *AdaptThink* via existing difficulty annotation on MATH Hendrycks et al. (2021) in Appendix Section B. We also exclude the impact of problem diversity on SFT performance in Appendix Section C. These experiments confirm that *AdaptThink* serves as an effective identifier for identifying high-value difficult problems, and that such problems are significantly more beneficial than simple ones in improving model performance.

### 3.2 Difficult Problem Generator

Building upon the identified difficult problem set $\mathcal{P}_{\text{Diff}}$ from $\mathcal{D}_{\text{Diff}}$ in Section 3.1, we train a dedicated difficult problem generator, denoted as DiffGen-8B, following a similar methodology to ScaleQuest (Ding et al., 2024b). The rationale for training exclusively on difficult problem sets derives from Section 3.1.2, which demonstrates that difficult problems are more effective than simple ones in improving model performance.

For each problem $x = (x_1, x_2, \ldots, x_n)$ in $\mathcal{P}_{\text{Diff}}$, where $x_i$ denotes the $i_{th}$ token of the problem, we construct the input by prepending a standard instruction prefix $\mathcal{I} = \texttt{<|im\_start|>user\textbackslash n}$. Distinct from conventional instruction tuning—where the loss is often computed over solution tokens—our training loss is only applied to the problem tokens $x_i$ without solution as input. The training objective for DiffGen-8B is the standard cross-entropy loss for language modeling:

$$\mathcal{L}_{\text{CE}}(\theta) = -\frac{1}{n} \sum_{i=1}^{n} \log P(x_i | \mathcal{I}, x_1, \ldots, x_{i-1}). \tag{3}$$

This design encourages DiffGen-8B to capture the distributional patterns inherent in challenging mathematical problems. Importantly, the goal is not to optimize for problem solving, but rather to enable the model to generate new problems of comparable complexity.

Given the instruction prefix $\mathcal{I}$, the trained generator DiffGen-8B can produce a large number of new difficult problems by adjusting decoding parameters such as temperature and top-p. The resulting collection of generated problems constitutes our problem set, denoted as $\mathcal{P}_{\text{DiffGen}}$.

### 3.3 Solution Distillation and Filtration

After generating the problem set $\mathcal{P}_{\text{DiffGen}}$, we re-evaluate their difficulty using the methodology introduced in Section 3.1. The validation shows that about 88% of the generated problems are classified as difficult, suggesting that DiffGen-8B effectively captures the distributional characteristics of challenging problems. Since assessing the mathematical correctness and solvability of generated problems remains a highly non-trivial task, we leave this aspect as future work and focus instead on ensuring the quality of distilled solutions.

For each problem in $\mathcal{P}_{\text{DiffGen}}$, we utilize a strong teacher model to distill its corresponding solution, resulting in $\hat{\mathcal{D}}_{\text{DiffGen}}$. Upon obtaining these solutions, we perform a two-stage filtration process: rule and model filtration. The initial rule filtering stage removes solutions with common undesirable traits. This includes cases with extensive repetition (20-token n-gram occurring more than 20 times) or overly verbose reasoning (total length exceeds 32,768 tokens or whose outputs do not contain the required `</think>` tag) that prevents the final answer from being clearly encapsulated within `\boxed{}`. The model-based filtering step further refines the dataset by discarding problems that the base model already solves consistently. Specifically, if the base model's predicted answer consistently matches the teacher-provided answer, the problem is treated as uninformative for further training and removed. This criterion identifies problems on which the base model and the teacher model do not provide meaningful disagreement. In total, we filter out approximately 43% of the initial samples and obtain the final problem-solution dataset $\mathcal{D}_{\text{DiffGen}}$.

## 4 Experiments

### 4.1 Experimental Setup

**Initial Dataset $\mathcal{D}$.** We utilize the mathematical domain subset of the AM-Qwen3-Distilled dataset[2] as our initial dataset $\mathcal{D}$, which is well-regarded for its high quality and has demonstrated effectiveness in training mathematical reasoning models. Its problem set $\mathcal{P}$ is a compilation of several prominent sub-datasets, including DeepMath-103K (He et al., 2025b), OpenR1-Math-220K (Face, 2025), OpenMathReasoning (Moshkov et al., 2025), and NuminaMath (Li et al., 2024b), among others. Subsequently, $\mathcal{P}$ undergoes rigorous deduplication, rule filtering, and decontamination concerning downstream tasks. The original solutions in $\mathcal{D}$ are distilled from Qwen3-235B-A22B (Yang et al., 2025). This distillation process is iteratively repeated until a correct solution is obtained. Further filtering is also conducted based on metrics such as perplexity and Ngram scores (Tian et al., 2025). This multi-stage processing results in a final curated dataset comprising 558K data instances.

**Training of DiffGen-8B.** Following the identification process described in Section 3.1, 192K problems from $\mathcal{P}$ are classified as difficult, denoted as $\mathcal{P}_{\text{Diff}}$. We train DiffGen-8B based on the Qwen3-8B-Base (Yang et al., 2025) model. The training configuration consists of a batch size of 128, a maximum sequence length of 1024 tokens, a learning rate of 5e-5, and a total of 1 epoch. We employ 10% warmup steps with a linear decay learning rate schedule. We use LLaMA-Factory (Zheng et al., 2024) as our training framework.

**Construction of $\mathcal{D}_{\text{DiffGen}}$.** Upon completion of training, we use DiffGen-8B to generate the $\mathcal{P}_{\text{DiffGen}}$. Generation parameters are set to a temperature of 1.0, a top-p value of 0.95, and a top-k value of 20. We utilize the Qwen3-8B model (Yang et al., 2025) as teacher model to generate long CoT solutions for the problems within $\mathcal{P}_{\text{DiffGen}}$ in "Thinking" mode with a temperature of 0.6, a top-p value of 0.95, and a top-k value of 20, resulting in $\hat{\mathcal{D}}_{\text{DiffGen}}$. These generated solutions then undergo the filtration process detailed in Section 3.3. For the model filtering stage, we specifically employ the Qwen2.5-Math-7B-Instruct (Yang et al., 2024b) model, given that this model also serves as the base model for our ScaleDiff. This comprehensive process yields our generated dataset, denoted as $\mathcal{D}_{\text{DiffGen}}$, comprising 1.15M problem-solution pairs. By augmenting $\mathcal{D}$ with $\mathcal{D}_{\text{DiffGen}}$, we get the final $\mathcal{D}_{Final}$ (ScaleDiff-Math), comprising 1.7M problem-solution pairs.

---

[2]`https://huggingface.co/datasets/a-m-team/AM-Qwen3-Distilled`

| Model | AIME'24 avg@10 | AIME'25 avg@10 | HMMT-Feb'25 avg@10 | BRUMO'25 avg@10 | MATH500 avg@3 | AVG |
|---|---|---|---|---|---|---|
| Qwen2.5-7B-Instruct (Yang et al., 2024a) | $11.3_{\pm 5.4}$ | $11.0_{\pm 5.2}$ | $2.7_{\pm 2.0}$ | $22.3_{\pm 3.7}$ | $77.5_{\pm 1.0}$ | 22.6 |
| Qwen2.5-Math-7B-Instruct (Yang et al., 2024c) | $11.3_{\pm 2.7}$ | $11.3_{\pm 3.1}$ | $2.0_{\pm 1.6}$ | $18.0_{\pm 6.0}$ | $82.7_{\pm 0.2}$ | 22.8 |
| DeepSeek-R1-Distill-Qwen-7B (DeepSeek-AI, 2025) | $53.0_{\pm 5.3}$ | $41.7_{\pm 6.5}$ | $25.0_{\pm 3.7}$ | $54.7_{\pm 7.2}$ | $93.7_{\pm 0.4}$ | 51.6 |
| Qwen3-8B (Yang et al., 2025) | $75.0_{\pm 4.5}$ | $64.7_{\pm 6.4}$ | $44.0_{\pm 4.4}$ | $68.0_{\pm 2.7}$ | $96.8_{\pm 0.3}$ | 68.9 |
| **RL** | | | | | | |
| LIMR-7B (Li et al., 2025a) | $33.3_{\pm 4.5}$ | $7.3_{\pm 3.3}$ | $0.7_{\pm 1.3}$ | $20.3_{\pm 4.3}$ | $77.4_{\pm 0.6}$ | 24.4 |
| Oat-Zero-7B (Liu et al., 2025a) | $30.0_{\pm 4.5}$ | $11.0_{\pm 4.0}$ | $4.0_{\pm 2.9}$ | $22.0_{\pm 3.7}$ | $79.4_{\pm 0.3}$ | 26.2 |
| Open-Reasoner-Zero-7B (Hu et al., 2025) | $17.0_{\pm 3.5}$ | $17.0_{\pm 3.1}$ | $3.0_{\pm 2.3}$ | $29.3_{\pm 2.9}$ | $82.4_{\pm 1.3}$ | 27.6 |
| AReaL-boba-RL-7B (Fu et al., 2025) | $58.0_{\pm 4.8}$ | $43.0_{\pm 4.8}$ | $25.3_{\pm 4.0}$ | $56.3_{\pm 5.8}$ | $93.2_{\pm 0.6}$ | 53.1 |
| Skywork-OR1-Math-7B (He et al., 2025a) | $59.7_{\pm 3.8}$ | $49.7_{\pm 5.0}$ | $30.3_{\pm 4.8}$ | $61.7_{\pm 4.5}$ | $95.3_{\pm 0.1}$ | 57.7 |
| Skywork-OR1-7B (He et al., 2025a) | $61.5_{\pm 4.2}$ | $50.3_{\pm 5.5}$ | $28.0_{\pm 5.0}$ | $63.7_{\pm 6.0}$ | $95.9_{\pm 0.2}$ | 58.3 |
| MiMo-7B-RL (Xiaomi et al., 2025) | $68.3_{\pm 4.3}$ | $59.0_{\pm 5.0}$ | $38.3_{\pm 4.8}$ | $64.3_{\pm 2.6}$ | $95.6_{\pm 0.4}$ | 64.1 |
| **SFT** | | | | | | |
| OpenThinker-7B (Team, 2025) | $28.0_{\pm 4.3}$ | $25.7_{\pm 4.7}$ | $18.0_{\pm 5.8}$ | $36.7_{\pm 4.7}$ | $87.9_{\pm 0.4}$ | 37.0 |
| OpenR1-Qwen-7B (Face, 2025) | $50.7_{\pm 5.1}$ | $36.3_{\pm 3.5}$ | $25.7_{\pm 3.0}$ | $55.7_{\pm 6.2}$ | $93.4_{\pm 0.7}$ | 49.7 |
| OpenThinker2-7B (Team, 2025) | $54.7_{\pm 7.6}$ | $38.0_{\pm 5.6}$ | $23.0_{\pm 4.1}$ | $54.7_{\pm 4.3}$ | $93.9_{\pm 0.4}$ | 50.4 |
| Light-R1-7B-DS (Wen et al., 2025b) | $55.3_{\pm 5.4}$ | $41.3_{\pm 2.7}$ | $26.7_{\pm 3.7}$ | $56.0_{\pm 4.9}$ | $94.0_{\pm 0.3}$ | 52.4 |
| MiMo-7B-SFT (Xiaomi et al., 2025) | $60.3_{\pm 6.0}$ | $44.3_{\pm 6.7}$ | $25.7_{\pm 4.5}$ | $50.7_{\pm 8.1}$ | $93.6_{\pm 0.2}$ | 53.2 |
| AceReason-Nemotron-7B (Chen et al., 2025) | $64.3_{\pm 2.6}$ | $50.3_{\pm 2.8}$ | $30.3_{\pm 3.5}$ | $63.7_{\pm 6.0}$ | $96.1_{\pm 0.4}$ | 59.2 |
| AM-Qwen3-Distilled-7B* (Tian et al., 2025) | $63.0_{\pm 3.5}$ | $51.7_{\pm 5.6}$ | $33.3_{\pm 5.8}$ | $60.7_{\pm 7.7}$ | $94.6_{\pm 0.4}$ | 59.2 |
| AM-Thinking-v1-Distilled-7B* (Tian et al., 2025) | $62.0_{\pm 5.8}$ | $50.0_{\pm 3.3}$ | $42.3_{\pm 4.0}$ | $62.7_{\pm 3.9}$ | $94.9_{\pm 0.7}$ | 60.3 |
| OpenThinker3-7B (Guha et al., 2025) | $66.3_{\pm 4.3}$ | $57.3_{\pm 5.5}$ | $36.0_{\pm 3.9}$ | $67.7_{\pm 3.0}$ | $95.8_{\pm 0.4}$ | 63.4 |
| OpenMath-Nemotron-7B (Moshkov et al., 2025) | $73.7_{\pm 4.1}$ | $60.7_{\pm 4.7}$ | $43.0_{\pm 5.5}$ | $68.0_{\pm 6.2}$ | $95.2_{\pm 0.3}$ | 66.9 |
| ScaleDiff-7B | $73.0_{\pm 5.0}$ | $58.7_{\pm 8.2}$ | $43.3_{\pm 4.2}$ | $66.7_{\pm 2.7}$ | $95.2_{\pm 0.3}$ | 65.9 |

Table 3: Pass@1 accuracy (mean $\pm$ std) comparison of different LRMs on AIME'24, AIME'25, HMMT-Feb'25, BRUMO'25, and MATH500 benchmarks with multiple runs. The baseline results are sorted by the average performance. * denotes results from our evaluation of the Qwen2.5-Math-7B-Instruct model trained by us on the corresponding dataset. The rows highlighted in gray correspond to the source data $\mathcal{D}$ used for the ScaleDiff augmentation.

**Training of ScaleDiff.** As described above, in SFT, ScaleDiff model is initialized from Qwen2.5-Math-7B-Instruct (Yang et al., 2024b) model and trained on ScaleDiff-Math dataset. The batch size is set to 32, the maximum sequence length is 32,768 tokens, the training epoch is set to 3, with other training settings consistent with those employed for training DiffGen-8B. Due to the native context length limitation of the Qwen2.5-Math-7B-Instruct model to 4,096 tokens, we modify the rope_theta parameter from 10K to 300K to enable support for a maximum context length of 32,768 tokens, following the practice of OpenR1 (Face, 2025). The data template used for fine-tuning follows the default format of Qwen series.

**Evaluation.** To ensure robust and reproducible results, our evaluation adheres to the standardized framework and best practices outlined in (Hochlehnert et al., 2025). We assess the performance of our ScaleDiff model against relevant baselines on a comprehensive set of widely recognized mathematical reasoning benchmarks: AIME'24 (AI-MO), AIME'25 (Lin, 2025), HMMT Feb'25 (HMMT, 2025), BRUMO (BRUMO, 2025), and MATH500 (Lightman et al., 2023). Performance is primarily measured using the standard Pass@1 metric. To account for potential variability, especially on smaller benchmarks, all evaluation results are averaged over multiple random seeds. Specifically, we use 10 random seeds for AIME'24, AIME'25, HMMT-Feb'25, BRUMO'25, and 3 random seeds for MATH500. The maximum number of new tokens, temperature, and top-p are set to 32,768, 0.6, and 0.95, respectively. All evaluations are conducted using the LightEval framework (Fourrier et al., 2023) with a vLLM backend (Kwon et al., 2023).

**Baselines.** We mainly compare ScaleDiff with Qwen2.5-7B model series, including Qwen2.5-7B-Instruct (Yang et al., 2024a), Qwen2.5-Math-7B-Instruct (Yang et al., 2024c), DeepSeek-R1-Distill-Qwen-7B (Guo et al., 2025), as well as LRMs that have undergone further SFT or RL based on Qwen2.5-7B model series.

## 4.2 MAIN RESULTS

Our ScaleDiff demonstrates strong performance on both relatively simple benchmark MATH500 and more challenging benchmarks including AIME, HMMT-Feb'25, and BRUMO, achieving average accuracies that surpass many RL- or SFT-based strong LRMs, such as MiMo-7B-RL (Xiaomi et al., 2025), Light-R1-7B-DS (Wen et al., 2025b), AceReason-Nemotron-7B (Chen et al., 2025), and the recent OpenThinker3-7B (Guha et al., 2025). Unlike most of these baseline methods, which rely on

| Model | Size | AIME'24 avg@10 | AIME'25 avg@10 | HMMT-Feb'25 avg@10 | BRUMO'25 avg@10 | MATH500 avg@3 | AVG |
|---|---|---|---|---|---|---|---|
| ScaleDiff | 192K | $61.0_{\pm 5.2}$ | $52.0_{\pm 5.0}$ | $33.0_{\pm 3.5}$ | $57.7_{\pm 5.0}$ | $94.7_{\pm 0.1}$ | 58.4 |
| w/o Rule Filtration | 192K | $59.3_{\pm 6.5}$ | $52.3_{\pm 2.1}$ | $31.0_{\pm 4.7}$ | $59.3_{\pm 3.6}$ | $94.3_{\pm 0.2}$ | 58.1 |
| w/o Rule & Model Filtration | 192K | $59.0_{\pm 7.5}$ | $46.7_{\pm 7.3}$ | $29.3_{\pm 7.9}$ | $56.7_{\pm 4.9}$ | $93.3_{\pm 0.5}$ | 55.3 |
| w/o Filtration & Difficult | 192K | $47.7_{\pm 5.8}$ | $45.0_{\pm 6.2}$ | $25.0_{\pm 3.4}$ | $47.0_{\pm 3.5}$ | $92.5_{\pm 0.9}$ | 50.4 |

Table 4: Ablation Study on the effects of difficult problem selection and response filtration.

rejection sampling during solution distillation—sampling multiple candidate solutions and retaining only those matching the ground-truth answer—our approach samples a single response per problem. This eliminates the need for repeated sampling until the correct solution is found, resulting in significantly lower data generation cost. Although the training data may contain incorrect answers, the diverse reasoning traces they provide can still contribute to enhancing the model's reasoning ability. This observation is consistent with prior findings reported in (Toshniwal et al.; Su et al., 2025).

**Comparison with AM-Qwen3-Distilled-7B.** ScaleDiff achieves substantial improvements (11.3%) over AM-Qwen3-Distilled-7B (Tian et al., 2025), as ScaleDiff-7B can be viewed as a "hiking" version of AM-Qwen3-Distilled-7B. Here hiking refers to increasing both the overall difficulty and volume of the dataset through the ScaleDiff pipeline. We believe that such difficulty hiking is generally applicable when the original dataset maintains a balanced difficulty distribution.

**Comparison with teacher model Qwen3-8B.** Qwen3-8B is the teacher model for ScaleDiff-7B. From the results in Table 3, ScaleDiff-7B achieves 65.9% average accuracy, which closely approaches Qwen3-8B's 68.9%. The gap between the two models is thus relatively small overall, indicating that the distillation and difficulty-hiking pipeline successfully transfers much of the teacher's reasoning ability into the student model.

### 4.3 ABLATION STUDY

We further conduct an ablation study to investigate the contributions of different components in the ScaleDiff pipeline. Specifically, we focus on two key modules: (1) difficult problem identification and (2) response filtration.

To verify the effect of difficult problem identification, we remove both the identification and the subsequent filtration steps, and instead train the question generator directly on the original problem set $\mathcal{P}$. We then generate new problems from this generator, distill responses from the same teacher model, and fine-tune the same target model. To assess the effect of response filtration, we keep the difficult problem identification step but remove the rule filtration or remove both the rule and model filtration. For fair comparison, the total fine-tuning data size is fixed to 192K samples across all experiments. The results are summarized in Table 4, from which we can observe: (1) Removing response filtration degrades performance ($58.4 \rightarrow 55.3$ on average), showing that both rule and model filtering are important to eliminate noisy, repetitive, or low-value samples, with model filtering contribute more on the performance drop ($58.1 \rightarrow 55.3$). This ensures the fine-tuning dataset remains both high-quality and challenging. (2) Removing difficult problem identification further causes a notable drop in performance ($55.3 \rightarrow 50.4$), confirming that pre-filtering challenging problems before generator training yields more effective data for enhancing reasoning capabilities. Without this step, the generated dataset may contain a higher proportion of trivial problems, limiting SFT gains.

## 5 ANALYSIS

In this section, we present a series of analyses to investigate the impact of data scaling (Section 5.1), the effect of teacher model (Section 5.2), and the difficulty of generated problems (Section 5.3). Unless otherwise specified, all experiments are conducted on unfiltered solutions.

### 5.1 IMPACT OF DATA SCALING

To assess the impact of augmentation scale on downstream performance, we vary the size of the generated dataset and evaluate the model across 3 benchmarks. Figure 2 illustrates the effect of

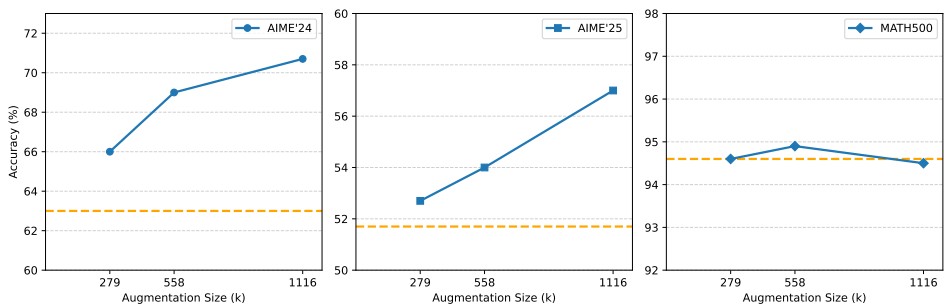

Figure 2: Accuracy scaling with the size of augmented data on AIME'24, AIME'25, and MATH500. The amount of augmented data is 1/2, 1, and 2 times the size of the original dataset.

| Teacher Model | Size | AIME'24 avg@10 | AIME'25 avg@10 | HMMT-Feb'25 avg@10 | BRUMO'25 avg@10 | MATH500 avg@3 | AVG |
|---|---|---|---|---|---|---|---|
| Qwen3-235B-A22B | 192K | $62.3_{\pm 5.0}$ | $44.3_{\pm 7.6}$ | $36.0_{\pm 5.7}$ | $59.0_{\pm 6.3}$ | $93.9_{\pm 1.2}$ | 56.6 |
| Qwen3-8B | 192K | $57.3_{\pm 5.3}$ | $50.0_{\pm 6.7}$ | $26.7_{\pm 6.3}$ | $56.3_{\pm 6.6}$ | $93.5_{\pm 1.0}$ | 55.6 |

Table 5: The effect of teacher model for solution distillation.

augmentation dataset size on model performance across three benchmarks: AIME'24, AIME'25, and MATH500. The yellow dashed line denotes the baseline results of AM-Qwen3-Distilled-7B without augmentation. From the figure, we observe a consistent performance improvement on the more challenging AIME'24 and AIME'25 benchmarks as the augmentation dataset size increases. Notably, even when the augmentation size reaches twice that of the original dataset, performance gains remain unsaturated, indicating the continued benefit of scaling difficult problems for enhancing complex reasoning. In contrast, for the relatively easier MATH500 benchmark, the augmentation provides no improvements, suggesting that additional difficult data contributes more significantly when the evaluation tasks themselves demand complex reasoning.

## 5.2 EFFECT OF TEACHER MODEL

The performance of different teacher models may vary, and consequently, the quality of their distilled responses can influence downstream results. In this section, we investigate the effect of the teacher model. In our pipeline, the original solutions for $\mathcal{D}_{\mathrm{Diff}}$ are distilled from Qwen3-235B-A22B. For each problem in $\mathcal{P}_{\mathrm{Diff}}$, we further distill three responses using Qwen3-8B. This is because Qwen3-8B occasionally fails on extremely difficult problems, and multiple attempts increase the chance of producing at least one correct solution. We then keep only one solution per problem—selecting a correct one if available, otherwise randomly choosing among the three. We then compare the results obtained from the two teacher models on this controlled dataset.

As shown in Table 5, we observe that using Qwen3-235B-A22B as the teacher model yields slightly better performance than Qwen3-8B, though the difference is not substantial. This finding partially aligns with prior observations in (Guha et al., 2025; Li et al., 2025b), which suggest that stronger-performing models are not necessarily better "teachers" because a noticeable gap often exists between large teacher models and smaller student models. These results corroborate our decision to adopt the smaller Qwen3-8B as a teacher model, demonstrating it to be a more cost-efficient choice.

## 5.3 DIFFICULTY OF GENERATED PROBLEMS

As described in Section 3.3, approximately 88% of the problems generated by DiffGen-8B are verified as difficult. To further investigate the characteristics of these problems, we analyze the distribution of response lengths across different datasets, namely $\mathcal{D}_{\mathrm{Simp}}$, $\mathcal{D}_{\mathrm{Diff}}$, and $\hat{\mathcal{D}}_{\mathrm{DiffGen}}$, as well as across different teacher models, Qwen3-235B-A22B and Qwen3-8B (use superscript L and S to represent them, respectively). The results are illustrated in Figure 3, from which several findings emerge.

(1) Comparing the distribution of $D_{\text{Simp}}^L$ (blue curve) with the others, we observe that the difficulty levels identified by *Adapt-Think* strongly correlate with response length: simple problems exhibit a sharp density peak at very short token lengths, reflecting their requirement for only brief solution traces, while difficult problems shift the distribution toward longer token lengths, consistent with the need for more elaborate reasoning chains. (2) Comparing the distribution of $D_{\text{Diff}}^L$ and $D_{\text{Diff}}^S$ (orange and green curves), we find

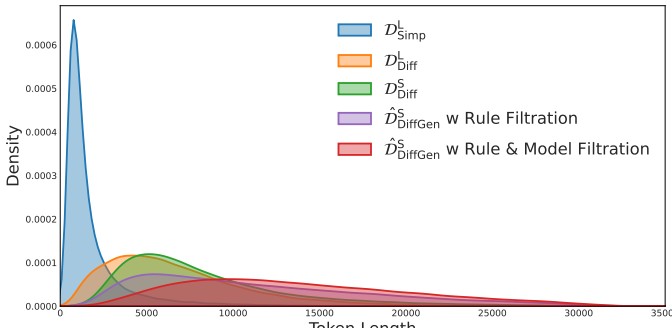

Figure 3: Distribution of solution lengths across datasets and teacher models. The superscript L denotes the use of the large-sized Qwen3-235B-A22B as the teacher model, whereas S indicates the use of the small-sized Qwen3-8B.

that the choice of teacher model from the same family (Qwen3) has little impact on the response length distribution for difficult problems, as both yield similar patterns. (3) Comparing the distribution of $D_{\text{Diff}}^L$ and $\hat{D}_{DiffGen}^S$ with rule and model filtration (orange and red curves) given (2), it becomes evident that generated problems tend to induce longer responses than original difficult problems, indicating higher intrinsic complexity. This observation is further corroborated by downstream results in Table 4 and 5: SFT on the 192K $D_{\text{Diff}}^L$ dataset yields an average performance of 56.6, whereas training on an equal amount of $\hat{D}_{DiffGen}^S$ with rule and model filtration achieves 58.4. (4) Finally, comparing the distribution of $\hat{D}_{DiffGen}^S$ with rule filtration and $\hat{D}_{DiffGen}^S$ with rule and model filtration (purple and red curves) shows that model filtration further refines the dataset by removing relatively easier problems, thereby retaining a subset of problems with greater difficulty and longer reasoning traces.

## 6 CONCLUSION

In this work, we introduce ScaleDiff, a simple yet effective pipeline for scaling the construction of difficult mathematical problems to enhance the complex reasoning abilities of LRMs. By leveraging *AdaptThink* as an efficient difficult problem identifier and training a dedicated generator (DiffGen-8B) to produce new difficult problems, we construct the ScaleDiff-Math dataset. Extensive experiments demonstrate that fine-tuning on this dataset yields substantial improvements over both strong SFT- and RL-based baselines across multiple mathematical reasoning benchmarks. Moreover, we observe a clear phenomenon that augmenting training data with increasing quantities of difficult problems consistently improves performance on challenging benchmarks, underscoring the value of difficulty-aware augmentation for advancing reasoning capabilities.

## REPRODUCIBILITY STATEMENT

Implementation details for training pipeline, datasets, and all hyperparameters are specified in Section 4.1. Our code is available at the anonymous repository `https://anonymous.4open.science/r/ScaleDiff-D053`.

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

## A  THE USE OF LARGE LANGUAGE MODELS (LLMS)

We used large language models (LLMs) to polish the manuscript, check clarity, and correct grammatical errors. The authors reviewed and remain responsible for all content.

## B  EFFECTIVENESS OF DIFFICULTY VIA ANNOTATIONS

To further examine whether *AdaptThink*'s difficulty recognition aligns with externally annotated difficulty levels, we analyze the relationship between *AdaptThink*'s identification and the official difficulty labels in the MATH Hendrycks et al. (2021) test set. The difficulty levels in MATH range from 1 to 5, following the annotation protocol of AoPS.

From the statistics in Table 6, we observe a clear monotonic trend: the proportion of "Thinking" mode predictions increases steadily as the annotated difficulty level increases. This indicates that *AdaptThink* adaptively chooses its reasoning mode in accordance with established difficulty labels, thereby providing additional evidence that the model's difficulty identification is consistent with externally defined problem difficulty.

## C  PROBLEM DIVERSITY

Since downstream performance may be influenced not only by problem difficulty but also by the diversity, we provide a quantitative analysis to ensure that diversity is not a confounding factor in our difficulty-based comparisons. To quantify the diversity of $\mathcal{P}_{\text{Simp}}$ and $\mathcal{P}_{\text{Diff}}$, we embed all problems using Qwen3-Embedding-0.6B Zhang et al. (2025b). We then compute the *global cosine diversity*, defined as:

$$\text{Div}_{\text{Global}} = 1 - \mathbb{E}\left[\cos(x, \bar{x})\right], \tag{4}$$

| Difficulty Level | Thinking Ratio (%) | NoThinking Ratio (%) |
|:---:|:---:|:---:|
| 1 | 2.7 | 97.3 |
| 2 | 4.6 | 95.4 |
| 3 | 10.9 | 89.1 |
| 4 | 21.7 | 78.3 |
| 5 | 41.5 | 58.5 |

Table 6: Thinking mode vs. NoThinking mode prediction ratios across annotated difficulty levels in the MATH test set.

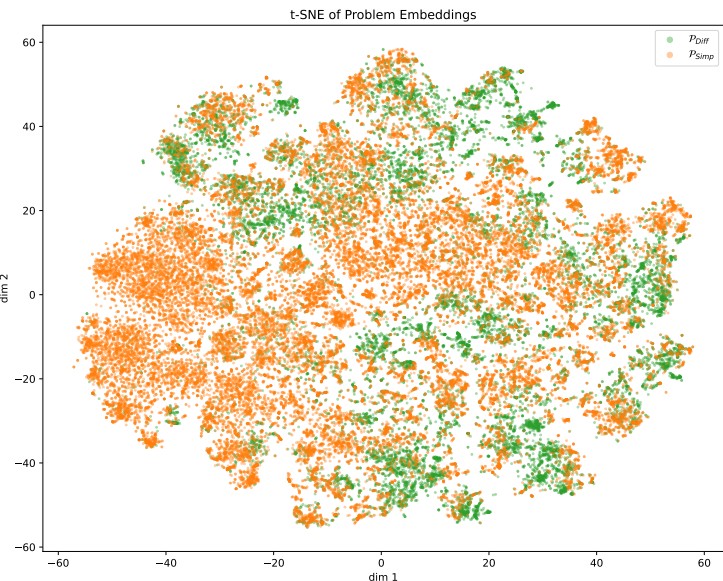

Figure 4: t-SNE visualization of downsampled (10%) $\mathcal{P}_{\text{Simp}}$ and $\mathcal{P}_{\text{Diff}}$.

where $\bar{x}$ denotes the mean embedding vector and all embeddings are $\ell_2$-normalized. Higher values of $\text{Div}_{\text{Global}}$ correspond to a greater spread of the distribution. The computed $\text{Div}_{\text{Global}}$ values for $\mathcal{P}_{\text{Simp}}$ and $\mathcal{P}_{\text{Diff}}$ are 0.495 and 0.473, respectively, indicating that diversity is adequately controlled in both cases. Furthermore, we randomly downsample 10% of $\mathcal{P}_{\text{Simp}}$ and $\mathcal{P}_{\text{Diff}}$ subsets and visualize their embeddings using t-SNE. As shown in Figure 4, $\mathcal{P}_{\text{Simp}}$ and $\mathcal{P}_{\text{Diff}}$ exhibit clearly distinguishable distributions. These results confirm that the simple subset does not suffer from low diversity, and thus diversity is not a confounding factor in our difficult analysis.

We further examine the relationship between diversity and downstream accuracy across all subsets. Notably, $\text{Div}_{\text{Global}}$ and performance do not exhibit a positive correlation; in some cases, the trend is even inverted. For example, $\mathcal{D}_{\text{Diff}}$ has the *lowest* $\text{Div}_{\text{Global}}$ among all subsets, yet delivers the second-highest accuracy (only below the full dataset $\mathcal{D}$). Conversely, although $\mathcal{D}_{\text{Simp}}$ and $\mathcal{D}_{\text{Diff}}$ have comparable diversity levels, their downstream performance differs substantially. We also note that the mixed subset $\mathcal{D}_{\text{Mix}}$ achieves performance comparable to the random subset $\mathcal{D}_{\text{Rand}}$, and when $\text{Div}_{\text{Global}}$ is controlled, the average performance increases with the proportion of difficult samples in the dataset. These results indicate that diversity alone cannot account for the observed performance disparities, and that problem difficulty remains a key determinant even after controlling for dataset size and diversity.

## D  GENERALIZATION TO OTHER MODEL FAMILY

To further examine the generality of ScaleDiff pipeline beyond the Qwen2.5-Math series, we expand our evaluation to include two additional model families: Llama3.1 Dubey et al. (2024) and DeepSeek-Math Shao et al. (2024). We evaluate two settings: (i) training solely on the difficult

subset $\mathcal{D}_{\text{Diff}}$ (192K examples), and (ii) training on the combined set $\mathcal{D}_{\text{Diff}} \cup \mathcal{D}_{\text{DiffGen}}$ (192K + 192K examples), in order to quantify the effect of ScaleDiff data augmentation.

| Model | Training Set | Size | AIME'24 avg@10 | AIME'25 avg@10 | HMMT-Feb'25 avg@10 | BRUMO'25 avg@10 | MATH500 avg@3 | AVG |
|---|---|---|---|---|---|---|---|---|
| Qwen2.5-Math-7B-Instruct | – | – | $11.3_{\pm2.7}$ | $11.3_{\pm3.1}$ | $2.0_{\pm1.6}$ | $18.0_{\pm6.0}$ | $82.7_{\pm0.2}$ | 22.8 |
| ScaleDiff-Qwen2.5-Math-7B-Instruct | $\mathcal{D}_{\text{Diff}}$ | 192K | $62.3_{\pm5.0}$ | $44.3_{\pm7.6}$ | $36.0_{\pm5.7}$ | $59.0_{\pm6.3}$ | $93.9_{\pm1.2}$ | 56.6 |
| ScaleDiff-Qwen2.5-Math-7B-Instruct | $\mathcal{D}_{\text{Diff}} \cup \mathcal{D}_{\text{DiffGen}}$ | 364K | $65.0_{\pm5.2}$ | $52.0_{\pm5.0}$ | $35.0_{\pm3.1}$ | $62.0_{\pm5.0}$ | $94.0_{\pm0.6}$ | 60.0 |
| Llama-3.1-8B-Instruct | – | – | $5.3_{\pm3.4}$ | $0.0_{\pm0.0}$ | $0.7_{\pm1.3}$ | $2.7_{\pm2.9}$ | $48.2_{\pm1.1}$ | 9.5 |
| ScaleDiff-Llama-3.1-8B-Instruct | $\mathcal{D}_{\text{Diff}}$ | 192K | $41.3_{\pm6.0}$ | $28.3_{\pm5.6}$ | $24.7_{\pm6.0}$ | $46.0_{\pm9.5}$ | $86.7_{\pm1.1}$ | 42.6 |
| ScaleDiff-Llama-3.1-8B-Instruct | $\mathcal{D}_{\text{Diff}} \cup \mathcal{D}_{\text{DiffGen}}$ | 364K | $46.3_{\pm6.4}$ | $41.3_{\pm6.4}$ | $28.3_{\pm4.5}$ | $49.7_{\pm5.9}$ | $88.2_{\pm1.6}$ | 49.2 |
| DeepSeek-Math-7B-Instruct | – | – | $0.7_{\pm1.3}$ | $1.0_{\pm1.5}$ | $0.0_{\pm0.0}$ | $1.7_{\pm2.2}$ | $44.7_{\pm0.8}$ | 8.2 |
| ScaleDiff-DeepSeek-Math-7B-Instruct | $\mathcal{D}_{\text{Diff}}$ | 192K | $27.3_{\pm6.8}$ | $25.7_{\pm4.7}$ | $16.0_{\pm3.6}$ | $33.7_{\pm4.6}$ | $83.1_{\pm1.1}$ | 35.3 |
| ScaleDiff-DeepSeek-Math-7B-Instruct | $\mathcal{D}_{\text{Diff}} \cup \mathcal{D}_{\text{DiffGen}}$ | 364K | $41.7_{\pm4.8}$ | $27.0_{\pm4.8}$ | $23.7_{\pm4.1}$ | $41.0_{\pm8.8}$ | $87.9_{\pm1.1}$ | 41.4 |

Table 7: Performance comparison across multiple model families trained under the ScaleDiff pipeline. Incorporating ScaleDiff generated data consistently improves performance across Qwen, Llama, and DeepSeek-Math models.

As summarized in Table 7, incorporating SCALEDIFF-generated data consistently improves performance across all three model families—Qwen, Llama, and DeepSeek-Math—despite their distinct architectures and pretraining pipelines. These results demonstrate that the benefits of ScaleDiff are not tied to Qwen2.5-Math's math-heavy specialization; instead, they transfer robustly to diverse model families, confirming the generality of our approach.

