# OpenReview forum: "ScaleDiff: Scaling Difficult Problems for Advanced Mathematical Reasoning"
_ICLR.cc/2026/Conference — ICLR 2026 Conference Withdrawn Submission_

### Official Review · Reviewer_KHT3 · 2025-10-24

**Soundness:** 2
**Presentation:** 2
**Contribution:** 2
**Rating:** 2
**Confidence:** 4

**Summary:**

This paper introduces ScaleDiff, a pipeline designed to address the challenge of creating difficult mathematical reasoning problems at scale for training Large Reasoning Models (LRMs). The core idea is to automate the generation of a large, high-quality dataset of challenging math problems to augment existing training data.

The ScaleDiff pipeline consists of four main stages:
- Identification: It first employs AdaptThink, an existing model, to efficiently identify and filter a subset of difficult problems from a large-scale, high-quality source dataset (AM-Qwen3-Distilled).
- Generation: A dedicated problem generator, named DiffGen-8B, is then trained exclusively on the text of these identified difficult problems to learn their underlying data distribution.
- Distillation: The trained DiffGen-8B generates a massive number of new, difficult problems. A strong teacher model (Qwen3-8B) is then used to generate step-by-step solutions (Chain-of-Thought) for these problems.
- Filtration: The resulting problem-solution pairs undergo a two-stage filtering process: a rule-based filter to remove malformed solutions and a model-based filter to discard problems that the target student model can already solve correctly.

The final augmented dataset is used to fine-tune a Qwen2.5-Math-7B model. The authors demonstrate through extensive experiments that their method yields substantial performance improvements across several challenging math benchmarks, including AIME'24, AIME'25, and HMMT-Feb'25, outperforming strong SFT and RL baselines. Ablation studies are also provided to validate the effectiveness of each key component in the pipeline.

**Strengths:**

- Addresses a Well-Defined and Important Problem: The paper tackles a significant bottleneck in advancing the reasoning capabilities of LRMs: the scarcity of high-difficulty, high-quality training data. The proposed goal of creating a scalable and automated data generation pipeline is highly relevant and valuable to the community.
- Systematic and Practical Pipeline: The ScaleDiff pipeline is a methodologically sound, end-to-end system. It is well-designed, practical, and demonstrates high efficiency by using a single forward pass for difficulty identification and a dedicated generator for scalable problem creation. This makes the approach a useful and reproducible blueprint for practitioners.
- Thorough Empirical Validation and Ablation Studies: The experimental evaluation is a standout feature of this work. The authors not only demonstrate strong final performance but also conduct detailed ablation studies (Table 3) that convincingly isolate and quantify the positive impact of the "difficult problem identification" and "response filtration" stages. This adds significant credibility to their claims.
- Strong and State-of-the-Art Results: The resulting ScaleDiff-7B model achieves impressive performance on a comprehensive set of mathematical reasoning benchmarks. It surpasses many larger models and strong, contemporary baselines, effectively demonstrating the real-world efficacy of the proposed data augmentation strategy.
- Provides Actionable Insights: The analysis section offers valuable insights, particularly the finding that a smaller teacher model (Qwen3-8B) can be nearly as effective for solution distillation as a much larger one from the same family. This has important practical implications for balancing cost and performance in model development.

**Weaknesses:**

- Limited Methodological Novelty: The paper's primary contribution lies in the effective integration and application of existing methods rather than the introduction of novel techniques. The difficulty identifier (AdaptThink) is an adopted tool, and the methodology for training the problem generator (DiffGen) is explicitly noted to follow a similar approach to ScaleQuest. While the resulting system is effective, the lack of foundational algorithmic novelty may be a concern from a research perspective.
- Absence of Evaluation on Generated Problem Diversity: The paper successfully shows that the generated problems are "difficult" (88% are classified as such by AdaptThink). However, there is no quantitative analysis of the diversity of these generated problems. It is unclear whether DiffGen is producing genuinely novel problems or merely creating syntactic paraphrases and variations of the problems in its training set (P_diff). This is a critical omission for evaluating the quality of a generative data pipeline.
- Potential for Dataset Contamination and Unclear Generalization: The pipeline starts with the AM-Qwen3-Distilled dataset, which aggregates several public math datasets. This introduces a risk that the training data distribution is already very close to that of the evaluation benchmarks (e.g., AIME), potentially inflating performance metrics. The paper would be strengthened by testing the generalization of its method, for instance, by training on a dataset from one domain (e.g., high school olympiads) and evaluating on a distinctly different one (e.g., college-level mathematics or SAT Math) to prove the pipeline's robustness.
- Ambiguity in the Contribution of the Model Filtering Step: The model filtering step, which discards problems the student model can already solve, is shown to be effective. However, its role could be interpreted as a highly targeted form of rejection sampling. It is not fully disentangled whether the performance gain comes from the sophisticated DiffGen training strategy or simply from having a very large pool of generated problems to apply this effective filtering technique on. The core contribution of DiffGen's training versus a simpler generation-and-filtering scheme is not entirely clear.
- Homogeneity of Teacher and Student Models in Analysis: The teacher model comparison in Section 5.2 uses Qwen3-235B and Qwen3-8B, which are from the same model family. It is highly likely that the 8B model is a distilled or aligned version of the 235B model, thus sharing very similar reasoning patterns. This could explain the minimal performance difference. The claim about the cost-effectiveness of smaller teachers would be far more compelling if validated with a heterogeneous teacher model from a different family (e.g., DeepSeek, Llama, Gemini) to show that the distillation process is robust across different model architectures and training methodologies.
- Lack of Comparison to Simpler Generation Baselines: The paper justifies the complexity of training a dedicated generator (DiffGen) but does not compare it against simpler, potent baselines. For example, applying advanced problem rephrasing or modification techniques (akin to MetaMath) directly to the identified difficult problem set (P_diff) could potentially achieve similar results in terms of difficulty and diversity, but with less computational overhead. Without this comparison, the necessity of the DiffGen component is not fully established.

**Questions:**

See Weakness.

---

> ### Author Response · Authors · 2025-11-17
>
> > W1: Limited Methodological Novelty: The paper's primary contribution lies in the effective integration and application of existing methods rather than the introduction of novel techniques. The difficulty identifier (AdaptThink) is an adopted tool, and the methodology for training the problem generator (DiffGen) is explicitly noted to follow a similar approach to ScaleQuest. While the resulting system is effective, the lack of foundational algorithmic novelty may be a concern from a research perspective.
> >
>
> Thank you for this valuable perspective. We would like to clarify that the core methodological contributions of ScaleDiff go beyond simply combining prior tools. Our work introduces several novel ideas that are essential to the system’s effectiveness:
>
> **(1) Difficulty as the primary organizing principle for synthetic math data.**
>
> Although prior work explores synthetic data generation, ScaleDiff systematically treats *difficulty* as a first-class signal, constructs a difficulty-stratified dataset at scale, and demonstrates that difficulty—not size or diversity—is the main driver of reasoning improvements.
>
> **(2) Difficulty-hiking generation with a dedicated weak-to-strong transfer pipeline.**
>
> DiffGen is not a direct replication of ScaleQuest. We show that training a generator *exclusively on filtered difficult instances* produces a qualitatively different distribution of problems, enabling effective transfer from an 8B teacher to smaller models. This “difficulty-hiking’’ mechanism is new.
>
> **(3) A new large-scale difficult math dataset.**
>
> We will release the dataset containing millions of difficult problems and solutions to the research community.
>
> > W2: Absence of Evaluation on Generated Problem Diversity: The paper successfully shows that the generated problems are "difficult" (88% are classified as such by AdaptThink). However, there is no quantitative analysis of the diversity of these generated problems. It is unclear whether DiffGen is producing genuinely novel problems or merely creating syntactic paraphrases and variations of the problems in its training set (P\_diff). This is a critical omission for evaluating the quality of a generative data pipeline.
> >
>
> We appreciate your insightful comments regarding the importance of evaluating problem diversity. To address this, we conducted a quantitative diversity analysis and compared ScaleDiff with an established data‐augmentation method MetaMath.
>
> Specifically, we embed all questions using **Qwen3-Embedding-0.6B** and measure diversity at the problem level.
>
> We compare:
>
> - **ScaleDiff dataset**: $\mathcal{D} \cup \mathcal{D}\_{DiffGen}$
> - **MetaMath Rephrased dataset**: $\mathcal{D}\_{\text{MetaMath}} \cup \mathcal{D}\_{\text{Rephrase}}$,
>
> Where  $\mathcal{D}\_{\text{MetaMath}}$ = GSM8K ∪ MATH and $\mathcal{D}\_{\text{Rephrase}}$ contains MetaMath-style rephrased questions.
>
> We compute the nearest-neighbor diversity for each dataset: $\text{Div}\_{\text{NN}} = \frac{1}{N}\sum\_{i=1}^N \left(1 - \max\_{j\neq i} x\_i^\top x\_j\right)$, where all embeddings are L2-normalized. This metric captures whether a dataset contains many near-duplicates (low diversity) or genuinely novel problems (higher diversity). Results show that $\text{Div}\_{\text{NN}}(\mathcal{D} \cup \mathcal{D}\_{\text{DiffGen}}) = \mathbf{0.116}$, while $\text{Div}\_{\text{NN}}(\mathcal{D}\_{\text{MetaMath}} \cup \mathcal{D}\_{\text{Rephrase}}) = \mathbf{0.011}$
>
> The rephrase-based augmented dataset exhibits extremely low nearest-neighbor diversity, consistent with the intuition that paraphrasing yields problems that are semantically similar to existing ones.
>
> In contrast, **ScaleDiff increases diversity by more than an order of magnitude**,  indicating that DiffGen produces genuinely novel and distributionally spread-out problems, rather than syntactic paraphrases of training items.

---

> ### Author Response · Authors · 2025-11-17
>
> > W3: Potential for Dataset Contamination and Unclear Generalization: The pipeline starts with the AM-Qwen3-Distilled dataset, which aggregates several public math datasets. This introduces a risk that the training data distribution is already very close to that of the evaluation benchmarks (e.g., AIME), potentially inflating performance metrics. The paper would be strengthened by testing the generalization of its method, for instance, by training on a dataset from one domain (e.g., high school olympiads) and evaluating on a distinctly different one (e.g., college-level mathematics or SAT Math) to prove the pipeline's robustness.
> >
>
> Our goal in this paper is to improve **general mathematical reasoning ability** rather than cross-domain generalization.
>
> The following two points ensure the different data distributions between training data and evaluation benchmarks.
>
> **(1) Strict decontamination of the seed dataset.**
>
> The AM-Qwen3-Distilled dataset undergoes rigorous filtering prior to use:
>
> - **Exact-match filtering** removes any instance identical to benchmark questions.
> - **Semantic filtering** embeds each problem using **bge-m3**, and removes items whose similarity to evaluation questions exceeds **0.9**.
>
> DiffGen is trained *exclusively* on this decontaminated corpus.
>
> **(2) Decontamination of generated problems.**
>
> We further run **exact-match checks** against all evaluation benchmarks for the generated problems and find **no overlaps**, confirming that DiffGen does not introduce test leakage.
>
> > W4: Ambiguity in the Contribution of the Model Filtering Step: The model filtering step, which discards problems the student model can already solve, is shown to be effective. However, its role could be interpreted as a highly targeted form of rejection sampling. It is not fully disentangled whether the performance gain comes from the sophisticated DiffGen training strategy or simply from having a very large pool of generated problems to apply this effective filtering technique on. The core contribution of DiffGen's training versus a simpler generation-and-filtering scheme is not entirely clear.
> >
>
> Thank you for the insightful question. We agree that it is important to disentangle the contribution of **DiffGen training** from that of **model-based filtering**.  As shown in the ablation study, removing the filtering step do lead to a performance drop, which raises the concern about the true source of the ScaleDiff performance gains.
>
> To further address your concern, we extend the ablation to the **full unfiltered dataset** . As shown in the following Table, removing the filtering step results in **slightly lower overall performance**, but it remains competitive with most strong baselines like MiMo-7B-RL [1] and OpenThinker3-7B [2]. This indicates that **DiffGen's training is the primary source of the performance improvements**, producing difficult problems that remain effective even without filtering, and **model-based filtering acts as a complementary refinement step**, improving data efficiency but not substituting for DiffGen’s targeted difficulty-hiking generation.
>
> |  | AIME'24 (avg@10) | AIME'25 (avg@10) | HMMT-Feb'25 (avg@10) | BRUMO'25 (avg@10) | MATH500 (avg@3) | AVG |
> | --- | --- | --- | --- | --- | --- | --- |
> | MiMo-7B-RL | 68.3±4.3 | 59.0±5.0 | 38.3±4.8 | 64.3±2.6 | 95.6±0.4 | 64.1 |
> | OpenThinker3-7B | 66.3±4.3 | 57.3±5.5 | 36.0±3.9 | 67.7±3.0 | 94.8±0.4 | 63.2 |
> | ScaleDiff | 73.0±5.0 | 58.7±8.2 | 43.3±4.2 | 66.7±2.7 | 95.2±0.3 | 65.9 |
> | ScaleDiff (No Filtration) | 70.7±3.6 | 57.0±5.9 | 43.7±8.1 | 64.0±6.3 | 94.5±0.1 | 64.5 |

---

> ### Author Response · Authors · 2025-11-17
>
> > W5: Homogeneity of Teacher and Student Models in Analysis: The teacher model comparison in Section 5.2 uses Qwen3-235B and Qwen3-8B, which are from the same model family. It is highly likely that the 8B model is a distilled or aligned version of the 235B model, thus sharing very similar reasoning patterns. This could explain the minimal performance difference. The claim about the cost-effectiveness of smaller teachers would be far more compelling if validated with a heterogeneous teacher model from a different family (e.g., DeepSeek, Llama, Gemini) to show that the distillation process is robust across different model architectures and training methodologies.
> >
>
> We appreciate your perspective. We select Qwen3-8B as the teacher primarily because distilling on million-scale data with significantly larger models would be prohibitively expensive. While a stronger or cross-family teacher could further improve results, the central contribution of ScaleDiff is not tied to a specific teacher architecture but to validating the *difficulty-hiking pipeline* under a controlled and cost-efficient setting.
>
> To more directly assess robustness beyond the Qwen family, we keep the teacher model fixed (Qwen3-8B) and vary the student models across heterogeneous families. We use Qwen3-8B as the teacher because most competitive 7–8B math models are indeed Qwen-based, and it is difficult to find comparably strong alternatives outside this family.
>
> Specifically, despite the Qwen family, we select two more model families—**Llama** [3] and **DeepSeek-Math** [4]-as our student model. Due to time constraints of rebuttal, we evaluate two settings on a representative subset of our data: (i) training solely on $\mathcal{D}\_{Diff}$ (192K), and (ii) training on $\mathcal{D}\_{\text{Diff}} \cup \mathcal{D}\_{\text{DiffGen}}$ (192K+192K), to measure the effect of data augmentation.
>
> The results summarized in the table below consistently show that incorporating ScaleDiff-generated data yields improvements across these model families. This demonstrates that the effectiveness of ScaleDiff is not tied to the Qwen family or its math-oriented pre/post-training, but transfers robustly to diverse modal families.
>
> We have included these new results to Appendix Section D of the revised manuscript.
>
> | Model | Training Set | Training Set Size | AIME'24 (avg@10) | AIME'25 (avg@10) | HMMT-Feb'25 (avg@10) | BRUMO'25 (avg@10) | MATH500 (avg@3) | AVG |
> | --- | --- | --- | --- | --- | --- | --- | --- | --- |
> | Qwen2.5-Math-7B-Instruct | - | - | 11.3±2.7 | 11.3±3.1 | 2.0±1.6 | 18.0±6.0 | 82.7±0.2 | 22.8 |
> | ScaleDiff-Qwen2.5-Math-7B-Instruct | $\mathcal{D}\_{Diff}$ | 192K | 62.3±5.0 | 44.3±7.6 | 36.0±5.7 | 59.0±6.3 | 93.9±1.2 | 56.6 |
> | ScaleDiff-Qwen2.5-Math-7B-Instruct | $\mathcal{D}\_{\text{Diff}} \cup \mathcal{D}\_{\text{DiffGen}}$ | 364K | 65.0±5.2 | 52.0±5.0 | 35.0±3.1 | 62.0±5.0 | 94.0±0.6 | 60.0 |
> | Llama-3.1-8B-Instruct | - | - | 5.3±3.4 | 0.0±0.0 | 0.7±1.3 | 2.7±2.9 | 48.2±1.1 | 9.5 |
> | ScaleDiff-Llama-3.1-8B-Instruct | $\mathcal{D}\_{Diff}$ | 192K | 41.3±6.0 | 28.3±5.6 | 24.7±6.0 | 46.0±9.5 | 86.7±1.1 | 42.6 |
> | ScaleDiff-Llama-3.1-8B-Instruct | $\mathcal{D}\_{\text{Diff}} \cup \mathcal{D}\_{\text{DiffGen}}$ | 364K | 46.3±6.4 | 41.3±6.4 | 28.3±4.5 | 49.7±5.9 | 88.2±1.6 | 49.2 |
> | deepseek-math-7b-instruct | - | - | 0.7±1.3 | 1.0±1.5 | 0.0±0.0 | 1.7±2.2 | 44.7±0.8 | 8.2 |
> | ScaleDiff-deepseek-math-7b-instruct | $\mathcal{D}\_{Diff}$ | 192K | 27.3±6.8 | 25.7±4.7 | 16.0±3.6 | 33.7±4.6 | 83.1±1.1 | 35.3 |
> | ScaleDiff-deepseek-math-7b-instruct | $\mathcal{D}\_{\text{Diff}} \cup \mathcal{D}\_{\text{DiffGen}}$ | 364K | 41.7±4.8 | 27.0±4.8 | 23.7±4.1 | 41.0±8.8 | 87.9±1.1 | 41.4 |

---

> ### Author Response · Authors · 2025-11-17
>
> > W6: Lack of Comparison to Simpler Generation Baselines: The paper justifies the complexity of training a dedicated generator (DiffGen) but does not compare it against simpler, potent baselines. For example, applying advanced problem rephrasing or modification techniques (akin to MetaMath) directly to the identified difficult problem set ($P\_{diff}$) could potentially achieve similar results in terms of difficulty and diversity, but with less computational overhead. Without this comparison, the necessity of the DiffGen component is not fully established.
> >
>
> Thank you for the helpful suggestion. We clarify that **DiffGen is intentionally designed to be lightweight**, not a heavy or expensive component. DiffGen is trained for only **one epoch** on the difficult problem set $P\_{\text{diff}}$, requires **no solution**, and therefore operates under a low maximum sequence length. Once trained, DiffGen can sample large quantities of diverse hard problems extremely efficiently, which is essential for scaling.
>
> Following your suggestion, we additionally compared DiffGen with the **MetaMath [5] Rephrase Question** augmentation method applied directly to $P\_{\text{diff}}$. As shown in the following, the MetaMath-style rephrasing yields **only marginal improvements**.
>
> We believe this is because rephrasing alters surface form but **does not change the underlying reasoning structure** of the problem. Consequently, the augmented problems remain semantically close to the originals and do not meaningfully expand the reasoning space.
>
> In contrast, DiffGen leverages the broad pretraining of **Qwen3-8B-Base** and the difficulty-focused training signal from $P\_{\text{diff}}$ to generate **novel, more varied, and substantively harder** problems. This qualitative difference in the distribution of generated tasks leads to the substantial performance gains we observe.
>
> | Model | Training Set | Training Set Size | AIME'24 (avg@10) | AIME'25 (avg@10) | HMMT-Feb'25 (avg@10) | BRUMO'25 (avg@10) | MATH500 (avg@3) | AVG |
> | --- | --- | --- | --- | --- | --- | --- | --- | --- |
> | ScaleDiff | $\mathcal{D}\_{Diff}$ | 192K | 62.3±5.0 | 44.3±7.6 | 36.0±5.7 | 59.0±6.3 | 93.9±1.2 | 56.6 |
> | ScaleDiff | $\mathcal{D}\_{\text{Diff}} \cup \mathcal{D}\_{\text{DiffGen}}$ | 364K | 65.0±5.2 | 52.0±5.0 | 35.0±3.1 | 62.0±5.0 | 94.0±0.6 | 60.0 |
> | MetaMath | $\mathcal{D}\_{\text{Diff}} \cup \mathcal{D}\_{\text{MetaMath}}$ | 364K | 62.5±6.5 | 45.0±2.1 | 35.7±4.7 | 59.3±3.6 | 94.3±0.2 | 57.0 |
>
> **References**
>
> [1] Xiaomi, L. L. M., et al. "MiMo: Unlocking the Reasoning Potential of Language Model--From Pretraining to Posttraining." *arXiv preprint arXiv:2505.07608* (2025).
>
> [2] Guha, Etash, et al. "OpenThoughts: Data Recipes for Reasoning Models." *arXiv preprint arXiv:2506.04178* (2025).
>
> [3] https://ai.meta.com/blog/meta-llama-3-1/
>
> [4] Shao, Zhihong, et al. "Deepseekmath: Pushing the limits of mathematical reasoning in open language models." *arXiv preprint arXiv:2402.03300* (2024).
>
> [5] Yu, Longhui, et al. "MetaMath: Bootstrap Your Own Mathematical Questions for Large Language Models." *The Twelfth International Conference on Learning Representations*.

---

> ### Author Response · Authors · 2025-11-26
>
> Dear Reviewer KHT3,
>
> Thank you again for your time and constructive feedback.
>
> We have posted a detailed response to your review comments a few days ago, addressing your concerns one by one. We would firmly appreciate it if you could let us know whether our response has sufficiently addressed your concerns or if there are any remaining questions we can clarify before the discussion period ends.
>
> Best regards, The Authors

---

### Official Review · Reviewer_ftno · 2025-11-01

**Soundness:** 2
**Presentation:** 3
**Contribution:** 3
**Rating:** 6
**Confidence:** 4

**Summary:**

This paper proposes a new framework, ScaleDiff, to efficiently generate and utilize difficult mathematical problems for training Large Reasoning Models. The method first identifies challenging problems from existing datasets using an adaptive model called AdaptThink, which distinguishes between “thinking” and “no-thinking” modes to infer difficulty with just one forward pass. These difficult examples are then used to train a specialized generator, DiffGen-8B, that produces new hard problems at scale without costly manual prompting. Solutions are distilled from a smaller, cost-efficient teacher model (Qwen3-8B) and filtered for quality, forming the ScaleDiff-Math dataset. Fine-tuning models on this dataset yields significant performance gains—an average accuracy of 65.9% across major math reasoning benchmarks, outperforming strong baselines like OpenThinker3. The paper also demonstrates a clear scaling trend: increasing the quantity of difficult problems systematically enhances reasoning performance, showing that difficulty-aware data generation is a scalable path to stronger reasoning models.

**Strengths:**

- Efficient difficulty identification via AdaptThink: a single forward pass “Thinking/NoThinking” switch replaces fail-rate loops and brittle LLM-as-judge prompts, with a precise operational criterion (first token).
- Large dataset: ScaleDiff-Math combines original data with 1.15M generated pairs to reach ~1.7M examples—substantial scale for open research.
- DiffGen-8B was trained exclusively on the subset of difficult problems identified by their pipeline. This design intentionally separates the task of problem generation from that of solution synthesis, allowing the model to specialize in producing challenging questions.
- Experiments showed a significant gain from performing SFT on ScaleDiff.

**Weaknesses:**

- Difficulty is defined by AdaptThink’s first-token decision, making the identifier inherently model-dependent and calibration-dependent, yet the paper provides no robustness checks (e.g., alternative identifiers, threshold sensitivity, cross-model agreement).
-  The generation stage optimizes only over problem tokens (no solution context), which can drift toward ill-posed or underspecified questions. Yet the paper seemed only to mention filtering out easy questions. Also, the cost-saving choice to distill a single solution per problem might cause some noise (e.g., incorrect solution or problematic reasoning).
- The claim "demonstrating that our pipeline can effectively transfer advanced reasoning capabilities without relying on larger, more expensive teacher models" sounds overstated to me. The teacher model (Qwen3-8B) is substantially stronger than the student model (Qwen2.5-Math-7B-Instruct), and the paper uses a 235B model for solution generation.

**Questions:**

- Have you tested the model on easier datasets like math500 and gsm8k? Disirably, ScaleDiff-7B should outperform baselines not only on the hardest datasets but rather on a whole spectrum of benchmarks.
- Have you conducted a decontamination or leakage check on the ScaleDiff dataset?
- Have you used ScaleDiff to train Qwen3-8B itself? The contribution would seem more compelling to me if the model (Qwen3-8B) could self-improve by generating harder data, since the whole data generation pipeline is somewhat expensive (using a 235B model).
- The paper uses an off-the-shelf model, AdaptThink, to classify simple/difficult questions. I would suggest an ablation comparing the current method with the existing ones. E.g., this paper: He, Yinghui, et al. "AdaptMI: Adaptive Skill-based In-context Math Instruction for Small Language Models." arXiv preprint arXiv:2505.00147 (2025).

---

> ### Author Response · Authors · 2025-11-17
>
> Thanks for your insightful review. We provide the following responses to address your concerns.
> > W1: Difficulty is defined by AdaptThink’s first-token decision, making the identifier inherently model-dependent and calibration-dependent, yet the paper provides no robustness checks (e.g., alternative identifiers, threshold sensitivity, cross-model agreement).
> >
> >
> > Q4: The paper uses an off-the-shelf model, AdaptThink, to classify simple/difficult questions. I would suggest an ablation comparing the current method with the existing ones.
> >
>
> Thanks for your suggestion. We further validate the effectiveness of AdaptThink's first-token decision via these 2 additional experiments.
>
> **(1) Additional validation using external difficulty annotations.**
>
> We further analyze the relationship between AdaptThink’s mode switch and difficulty levels labeled in the test set of MATH [1].
>
> The difficulty levels of MATH [1] range from 1 to 5, which is labeled according to [AoPS](https://artofproblemsolving.com).
>
> From the statistics in following table, we observe a clear monotonic trend that the proportion of Thinking mode predictions **increases steadily** with the annotated difficulty level. This indicates that AdaptThink **adaptively selects** its reasoning mode in alignment with established difficulty labels, providing additional support beyond downstream SFT metrics.
>
> | Difficult Level | Thinking Ratio | NoThinking Ratio |
> | --- | --- | --- |
> | 1 | 2.7 | 97.3 |
> | 2 | 4.6 | 95.4 |
> | 3 | 10.9 | 89.1 |
> | 4 | 21.7 | 78.3 |
> | 5 | 41.5 | 58.5 |
>
> **(2) Direct difficulty measurement via Pass@k.**
>
> Pass@k is commonly regarded as an upper bound on a model's reasoning capability. We further assess difficulty independently of SFT by evaluating Pass@3 directly on the problems. Specifically, we randomly sample 2K problems from $\mathcal{D}\_{Simp}$ and $\mathcal{D}\_{Diff}$ and test **three models** (two CoT types).
>
> As shown in the following table:
>
> - All models exhibit substantially lower Pass@3 on the Difficult-2K subset.
> - The gap is especially large for Short-CoT models, which are more sensitive to reasoning difficulty.
>
> | Model | CoT Type | Simple 2K | Difficult 2K |
> | --- | --- | --- | --- |
> | Qwen2.5-Math-7B-Instruct | Short | 88.2 | 65.8 |
> | Qwen3-8B (NoThinking) | Short | 85.8 | 71.5 |
> | Qwen3-8B (Thinking) | Long | 91.1 | 86.0 |
> | DeepSeek-R1-Distill-Qwen-7B | Long | 93.4 | 85.0 |
>
> These new experiments support the effectiveness of AdaptThink's first-token decision for difficulty identification.
>
> For comparing AdaptThink with other methods, we want to clarify that in the ScaleDiff pipeline, our focus is primarily on the *overall effectiveness* of difficulty-guided data generation rather than on benchmarking difficulty-identification methods themselves. We adopt AdaptThink because it offers a **cost-efficient** way for separating simple vs. difficult problems, which makes it practical at scale.
>
> While many alternative approaches exist—e.g., LLM-as-a-judge, fail-rate, response-length heuristics, consistency ratios, or reward-model scoring—most of these need solution generation and substantially higher computational cost. Moreover, the notion of “difficulty’’ in mathematical reasoning lacks a universally agreed-upon formal definition, making direct and fair comparison among these heterogeneous heuristics challenging. For these reasons, we did not include a systematic comparison of different difficulty classifiers in the paper.
>
> We have added these new analyses to Section 3.1.1 and Appendix Section B of the revised manuscript.

---

> ### Author Response · Authors · 2025-11-17
>
> > W2: The generation stage optimizes only over problem tokens (no solution context), which can drift toward ill-posed or underspecified questions. Yet the paper seemed only to mention filtering out easy questions. Also, the cost-saving choice to distill a single solution per problem might cause some noise (e.g., incorrect solution or problematic reasoning).
> >
>
> Thank you for the constructive feedback. Detecting ill-posed or underspecified questions is intrinsically challenging. In preliminary experiments, we applied an LLM-as-a-judge filter to a subset of the generated problems, but found that it provided **no reliable improvements**. Even with strong LLMs, accurately identifying subtle specification issues in difficult math problems remains difficult. For this reason, we omitted this step from the final pipeline.
>
> Regarding solution distillation, we also experimented with a **self-consistency–based filter**: for each generated problem, we sampled 8 candidate solutions and retained the reasoning trace associated with the majority answer. As shown in the following table, this procedure does not yield performance gains. We attribute this to the tendency of majority voting to reflect superficial agreement rather than deeper correctness on very challenging problems, resulting in limited benefit despite increased computational cost.
>
> Therefore, we adopted the more efficient **one-shot distillation without problem filtering** in the final system.
>
> |Model|AIME'24 (avg@10)|AIME'25 (avg@10)|HMMT-Feb'25 (avg@10)|BRUMO'25 (avg@10)|MATH500 (avg@3)|AVG|
> |---|---|---|---|---|---|---|
> |ScaleDiff|73.0±5.0|58.7±8.2|43.3±4.2|66.7±2.7|95.2±0.3|65.9|
> |ScaleDiff (Self-Consistency)|71.0±4.5|60.3±7.2|39.7±4.1|66.0±8.4|95.7±1.4|65.5|
>
> > W3: The claim "demonstrating that our pipeline can effectively transfer advanced reasoning capabilities without relying on larger, more expensive teacher models" sounds overstated to me. The teacher model (Qwen3-8B) is substantially stronger than the student model (Qwen2.5-Math-7B-Instruct), and the paper uses a 235B model for solution generation.
> >
>
> Our pipeline **does not** use Qwen3-235B-A22B for solution generation. Solution distillation for generated problems is performed with **Qwen3-8B**, which is a comparatively small model in the context of math-reasoning distillation pipelines.
>
> Despite relying only on an 8B-scale teacher, ScaleDiff achieves strong performance—surpassing several existing reasoning models that depend on **much larger** teacher models.
>
> > Q1: Have you tested the model on easier datasets like math500 and gsm8k? Disirably, ScaleDiff-7B should outperform baselines not only on the hardest datasets but rather on a whole spectrum of benchmarks.
> >
>
> Thank you for the question. Yes, we have evaluated ScaleDiff on easier benchmarks. As shown in Table 2, ScaleDiff-7B achieves strong performance on **MATH500 (avg@3)**, demonstrating that the benefits of our method are not restricted to only the hardest datasets. However, because MATH500 is relatively easy, the performance gap among different models is naturally smaller, leading to reduced discriminative power compared to challenging benchmarks.
>
> > Q2: Have you conducted a decontamination or leakage check on the ScaleDiff dataset?
> >
>
> Yes. The seed dataset (AM-Qwen3-Distilled) underwent **strict decontamination** using (i) exact-match filtering and (ii) semantic filtering, where each query is embedded using bge-m3 and any instance with similarity > 0.9 to evaluation benchmarks is removed. Since DiffGen is trained solely on this decontaminated problem set, it does not produce problems overly similar to test-set.
>
> For the generated problems, we additionally performed exact-match filtering against all evaluation benchmarks and found no overlaps.
>
> > Q3: Have you used ScaleDiff to train Qwen3-8B itself? The contribution would seem more compelling to me if the model (Qwen3-8B) could self-improve by generating harder data, since the whole data generation pipeline is somewhat expensive (using a 235B model).
> >
>
> We clarify that our pipeline **does not** use Qwen3-235B-A22B for solution generation; all solution distillation for generated problems is carried out using **Qwen3-8B** only, and the student model is Qwen2.5-Math-7B-Instruct.
>
> We have not explored using ScaleDiff-generated data to train Qwen3-8B itself. Such a setting falls under the broader direction of **model self-evolution**. This is an exciting research topic, but orthogonal to the focus of our work. Our primary goal in this paper is to study **knowledge transfer from a strong model to a weaker model** via difficulty-guided synthetic data generation. We therefore focus on the stronger-to-weaker distillation setting rather than self-improvement.
>
> **References**
>
> [1] Hendrycks, Dan, et al. "Measuring Mathematical Problem Solving With the MATH Dataset." *Thirty-fifth Conference on Neural Information Processing Systems Datasets and Benchmarks Track (Round 2)*.

---

> ### Author Response · Authors · 2025-11-26
>
> Dear Reviewer ftno,
>
> Thank you again for your time and constructive feedback.
>
> We have posted a detailed response to your review comments a few days ago, addressing your concerns one by one.
> We would firmly appreciate it if you could let us know whether our response has sufficiently addressed your concerns or if there are any remaining questions we can clarify before the discussion period ends.
>
> Best regards, The Authors

---

### Official Review · Reviewer_CauM · 2025-11-01

**Soundness:** 3
**Presentation:** 3
**Contribution:** 3
**Rating:** 6
**Confidence:** 3

**Summary:**

The authors proposed ScaleDiff, an efficient system to scale the creation of difficult mathematical problems for large reasoning models. The authors proposed to filter the hard problems with a low-parameter system that can efficiently work in one single forward pass. The effectiveness of this system is justified through a pre-experiment on the system. After verification of the evaluation function, a diffcult problem generator is trained and only the difficult generated problems are left. The benchmark results show that the models finetuned with the final augmented dataset outperforms existing baselines.

**Strengths:**

1. The paper is clearly written and presented.
2. The authors give clear clarification and intuition of why and how the system works.
3. The evaluation results show that the method is quite effective despite its simplicity.

**Weaknesses:**

The paper shows no significant weakness within its claim.

**Questions:**

1. The chapter "effectiveness of difficulty" seems to be problematic. The authors trained models on simple/hard divisions of the data with controlled number, and compared the metrics to that with full or random-selected dataset. The difficult subset seems to be effective given it providing similiar performance with smaller dataset size with full dataset. The simple subset, however, looks to me a too easy baseline. For example, what if we mix the simple subset with 20% data from the difficult subset? The peformance drop can happen simply because there is lack of diversity in simple subset, rather than the metric is really meaningful. I hope the authors could clarify this point.

---

> ### Author Response · Authors · 2025-11-17
>
> We appreciate your insightful comments regarding the potential role of **diversity** in our analysis of difficulty effectiveness.
>
> To quantify the diversity of the simple problem set $\mathcal{P}\_{Simp}$ and difficult problem set $\mathcal{P}\_{Diff}$, we embed all problems using Qwen3-Embedding-0.6B. Then we compute *global cosine diversity*, defined as  $\text{Div}\_{\text{Global}} = 1 - \mathbb{E}\left[\cos(x, \bar{x})\right]$, where $\bar{x}$ is the mean embedding vector and all embeddings all L2-normalized. $\text{Div}\_{\text{Global}}$ correlates positively with distributional spread.
>
> The $\text{Div}\_{\text{Global}}$ of $\mathcal{P}\_{Simp}$ and $\mathcal{P}\_{Diff}$ are 0.495 and 0.473, respectively, indicating that the **simple subset $\mathcal{P}\_{Simp}$ is slightly more diverse** than the difficult subset $\mathcal{P}\_{Diff}$. This suggests that the simple subset $\mathcal{P}\_{Simp}$ is not a low-diversity baseline.
>
> We also downsample 10% of $\mathcal{P}\_{Simp}$ and $\mathcal{P}\_{Diff}$ and visualize  their embeddings via t-SNE. From the t-SNE in Appendix Section C of revised manuscript, we can find that $\mathcal{P}\_{Simp}$ and $\mathcal{P}\_{Diff}$ both exhibit distinct distributions.
>
> Following your suggestion, we further construct a mixed dataset by augmenting the simple subset with **20%** of the difficult subset. This increased its diversity (0.495 → 0.497), still exceeding that of the $\mathcal{P}\_{Diff}$. However, as shown in the following table, the **average performance still lags behind** the model trained purely on $\mathcal{D}\_{\text{Diff}}$.
>
> These results indicate that **difficulty—not diversity—is the primary factor** driving the observed performance differences.
>
> | Dataset | Size | AIME'24 (avg@10) | AIME'25 (avg@10) | HMMT-Feb'25 (avg@10) | BRUMO'25 (avg@10) | MATH500 (avg@3) | AVG |
> |---| --- | --- | --- | --- | --- | --- | --- |
> |$\mathcal{D}$|558K|63.0±3.5|51.7±5.6|33.3±5.8|60.7±7.7|94.6±0.4|59.2|
> |$\mathcal{D}\_{Simp}$| 366K |43.7±5.3|38.7±4.5|27.0±5.3|53.7±6.0|91.3±0.4|48.9|
> |$\mathcal{D}\_{Simp}$|192K|40.7±4.9|33.7±2.8|24.0±3.6|48.3±7.8|90.4±0.7|45.1|
> |$\mathcal{D}\_{Simp}$ + 20% $\mathcal{D}\_{Diff}$|404K|53.7±4.3|41.3±7.0|30.0±4.7|55.3±6.2|93.5±0.3|52.5|
> |$\mathcal{D}\_{Rand}$|192K|54.3±5.0|42.0±5.2|31.3±4.8|57.0±4.6|93.2±0.4|53.3|
> |$\mathcal{D}\_{Diff}$|192K|62.3±5.0|44.3±7.6|36.0±5.7|59.0±6.3|93.9±1.2|56.6|
>
> We further examine the difficulty characteristics of the data through Pass@k performance and existing difficulty annotations.
>
> **(1) Direct difficulty measurement via Pass@k.**
>
> Pass@k is commonly regarded as an upper bound on a model’s reasoning capability. We further assess difficulty by evaluating Pass@k performance on the problems from $\mathcal{D}\_{\text{Simp}}$ and $\mathcal{D}\_{\text{Diff}}$.
>
> We randomly sample 2K problems from $\mathcal{D}\_{\text{Simp}}$ and $\mathcal{D}\_{\text{Diff}}$ and test **three models** with two CoT types: Qwen2.5-Math-7B-Instruct, Qwen3-8B under NoThinking and Thinking mode, and DeepSeek-R1-Distill-Qwen-7B. We set k=3 for all experiments.
>
> As shown in the following table, we can draw the following two conclusions.
>
> - All models exhibit substantially lower Pass@3 on the Difficult-2K subset.
> - The gap is especially large for Short-CoT models, which are more sensitive to reasoning difficulty.
>
> | Model | CoT Type | $\mathcal{D}\_{\text{Simp}}$-2K | $\mathcal{D}\_{\text{Diff}}$-2K |
> | --- | --- | --- | --- |
> |Qwen2.5-Math-7B-Instruct|Short|88.2|65.8|
> |Qwen3-8B (NoThinking)|Short|85.8|71.5|
> |Qwen3-8B (Thinking)|Long|91.1|86.0|
> |DeepSeek-R1-Distill-Qwen-7B|Long|93.4|85.0|
>
> **(2) Additional validation using external difficulty annotations.**
>
> We further analyze the relationship between AdaptThink’s mode switch and difficulty levels labeled in the test set of MATH [1].
>
> The difficulty levels of MATH range from 1 to 5, which is labeled according to [AoPS](https://artofproblemsolving.com).
>
> From the statistics in following table, we observe a clear monotonic trend that the proportion of Thinking mode predictions **increases steadily** with the annotated difficulty level. This indicates that AdaptThink **adaptively selects** its reasoning mode in alignment with established difficulty labels, providing additional support beyond downstream SFT metrics.
>
> |Difficult Level|Thinking Ratio|NoThinking Ratio|
> |---|---|---|
> |1|2.7|97.3|
> |2|4.6|95.4|
> |3|10.9|89.1|
> |4|21.7|78.3|
> |5|41.5|58.5|
>
> These new experiments support our claim that the AdaptThink Thinking/NoThinking switch provides a **meaningful signal of inherent question difficulty**.
>
> We have added these analyses, visualizations, and metrics to Section 3.1, Appendix Section B and C of the revised manuscript to clarify this point.
>
> **References**
>
> [1] Hendrycks, Dan, et al. "Measuring Mathematical Problem Solving With the MATH Dataset." *Thirty-fifth Conference on Neural Information Processing Systems Datasets and Benchmarks Track (Round 2)*.

---

> > ### Comment · Reviewer_CauM · 2025-11-17
> >
> > Thanks for your informative feedback. I think the authors have partially addressed my question. From the table provided, the performance of $\mathcal{D}_{Simp}$ + 20% $\mathcal{D}_{Diff}$ is equal to that of $\mathcal{D}_{Rand}$, which suggests the diversity also plays a vital role in the final performance. However, this should at least set a reasonable baseline to evaluate the effectiveness of your methods.

---

> > > ### Author Response · Authors · 2025-11-18
> > >
> > > We gratefully thank the reviewer again for the thoughtful follow-up.
> > >
> > > Following your suggestion, we have included the mixed subset  $\mathcal{D}\_{\text{Simp}}$ + 20% $\mathcal{D}\_{\text{Diff}}$ in **Table 1 of the revised manuscript** as a **diversity-aware baseline**, and add more discussions in **Appendix Section C** **of the revised manuscript.**
> > >
> > > We additionally report the ${\text{Div}}\_{\text{Global}}$ values and size of difficult samples for all subsets to make the diversity–performance relationship explicit.
> > >
> > > Our objective with these experiments is to clarify that, **while diversity indeed contributes to model performance, it is not the primary factor driving the large performance gap between $\mathcal{D}\_{\text{Simp}}$ and $\mathcal{D}\_{\text{Diff}}$.**
> > >
> > > As shown in the following table,
> > >
> > > (1) ${\text{Div}}\_{\text{Global}}$ and downstream accuracy are not positively correlated—in some cases, they even exhibit an inverse trend. For instance:
> > > - $\mathcal{D}\_{\text{Simp}}$ and $\mathcal{D}\_{\text{Diff}}$ have **similar** diversity values, but their performance differs **substantially**.
> > > - $\mathcal{D}\_{\text{Diff}}$ has the lowest ${\text{Div}}\_{\text{Global}}$ among all subsets, yet achieves the **second-best** performance (only below the full dataset $\mathcal{D}$).
> > >
> > > (2) **When $\text{Div}\_\text{Global}$ is controlled, the average performance increases with the proportion of difficult samples in the dataset.**
> > >
> > > | Dataset | ${Div}\_{Global}$ | Size | Difficult Size | AIME'24 (avg@10) | AIME'25 (avg@10) | HMMT-Feb'25 (avg@10) | BRUMO'25 (avg@10) | MATH500 (avg@3) | AVG |
> > > | --- | --- | --- | --- | --- | --- | --- | --- | --- | --- |
> > > | $\mathcal{D}$ | 0.497 | 558K | 192K | 63.0±3.5 | 51.7±5.6 | 33.3±5.8 | 60.7±7.7 | 94.6±0.4 | 59.2 |
> > > | $\mathcal{D}\_{Simp}$ | 0.495 | 366K | 0 | 43.7±5.3 | 38.7±4.5 | 27.0±5.3 | 53.7±6.0 | 91.3±0.4 | 48.9 |
> > > | $\mathcal{D}\_{Simp}$ | 0.476 | 192K | 0 | 40.7±4.9 | 33.7±2.8 | 24.0±3.6 | 48.3±7.8 | 90.4±0.7 | 45.1 |
> > > | $\mathcal{D}\_{Simp}$ + 20% $\mathcal{D}\_{Diff}$ | 0.497 | 404K | 38K | 53.7±4.3 | 41.3±7.0 | 30.0±4.7 | 55.3±6.2 | 93.5±0.3 | 52.5 |
> > > | $\mathcal{D}\_{Rand}$ | 0.497 | 192K | 66K | 54.3±5.0 | 42.0±5.2 | 31.3±4.8 | 57.0±4.6 | 93.2±0.4 | 53.3 |
> > > | $\mathcal{D}\_{Diff}$ | 0.473 | 192K | 192K | 62.3±5.0 | 44.3±7.6 | 36.0±5.7 | 59.0±6.3 | 93.9±1.2 | 56.6 |
> > >
> > > **We appreciate your suggestions and would be happy to address any further questions.**

---

> > > > ### Comment · Reviewer_CauM · 2025-11-18
> > > >
> > > > I agree it's a good idea to include ${Div}_{Global}$ and Difficult Size in your table and I appreciate that. This table clarifies my question.

---

### Official Review · Reviewer_s7ya · 2025-11-02

**Soundness:** 2
**Presentation:** 3
**Contribution:** 2
**Rating:** 2
**Confidence:** 5

**Summary:**

This work attempts to generate difficult synthetic problems meant to be used for training powerful mathematical reasoning problem. The approach leverages the AdaptThink model in order to filter out the difficult problems from a seed dataset which are then used for training synthetic problem generation model. After sampling new synthetic data from this problem generator, distilling its solutions and some filtering, the new synthetic problems are combined with the originally filtered seed dataset to form the final training data. Qwen2.5-Math-7B-Instruct trained on this final dataset shows improved average performance across 5 standardly used math reasoning benchmarks.

**Strengths:**

* The paper in general is clearly written and easy to follow.
* The idea of using dynamic thinking mode switching in AdaptThink as an indicator for difficulty is interesting
* I appreciate the the Abalysis in Section 5
* A good set of ablations has been included

**Weaknesses:**

I have several concerns about some of the claims made as well as the experimental design of this paper.

* The claim that the thinking vs non-thinking mode switching of AdaptThink can be used as a proxy to judge difficulty is not well substantiated. The authors demonstrate the effectiveness of different subsets of the dataset filtered by AdaptThink as SFT data in Table 1. However, downstream performance cannot be considered as an indicator for the difficulty itself, since it would depend on many factors other than the difficulty of the questions itself, such as but not limited to: diversity of the problems, quality of solution traces, etc. In general, I believe all the factors affecting the usefulness of a problem set as finetuning data are not well understood, thereby introducing confounders. A more direct way of substantiating the claim would be to measure pass@k of the filtered data subsets under multiple models.
* The approach does not involve any form of solution verification, leaving it as future work. However, I believe some basic form of solution verification is necessary to ensure some level of correctness, especially given that 88% of the generated problems are considered to be difficult. This could be something simple like self-consistency based verification.
* Sampling parameters for the solution distillation step are not mentioned?
* Sufficient details for the filtering process carried out after solution distillation are missing. How is verbosity / redundancy in the solutions detected? Do authors use LLM-as-a-judge?
* For model based-filtering, using agreement with the teacher model as a criteria for filtering presumes that all the solutions generated by the teacher model are correct, which is not guaranteed or even attempted to be ensured since there is no solution verification involved.
* I am skeptical of the sole use of Qwen2.5-7B-Math-Instruct as the base models given the recent results these models tend to improve even without data lacking any correct labels [1, 2] as a result of their heavy math oriented pre / post-training. Ideally, the effectiveness of the data should be demonstrated on multiple (atleast 3) base models, preferable across different model families such as Llama and Gemma family of models.
* Since this is an approach for synthetic data generation, the baselines should include other synthetic data gen methods. For eg., ScaleQuest [3] is an essential as well as strong baseline which needs to be included since it also involved training specialist synthetic data generation models. Other baselines include persona based synthetic data generation [4], DART-Math [5], MetaMath [1], etc. All these methods have publicly released datasets, making a data-sized match comparison possible.
* Section 4.2 - Comparison with the teacher model performance in not informative in my opinion since the performance would also depend on the base model (Qwen2.5-7B-Math-Instruct in this case). Finetuning a better model as the base can easily lead to performance exceeding that of the base model.
* Most of the results in Table 3 do not seems statistically significant given the small differences between means and high standard deviations. Thus, the conclusions made on the basis of this could be wrong.
* Section 5.3, point (3): Comparison of the response lengths of D^{L}_{Diff} and D^{S}_{DiffGen} is not a fair since they use two different solution generation models and the response lengths can vary across the two models, thus introducing a confounder.


### Other Comments (Not affecting my rating)
* Citations missing for AIME, MATH, BRUMO, HMMT in the intro.
* MATH500 citation is wrong. The correct citation should be:

Hunter Lightman, Vineet Kosaraju, Yuri Burda, Harrison Edwards, Bowen Baker, Teddy Lee, Jan Leike, John Schulman, Ilya Sutskever, and Karl Cobbe. Let’s verify step by step. In The Twelfth International Conference on Learning Representations, 2023.


### References
[1] https://arxiv.org/abs/2309.12284
[2] https://openreview.net/forum?id=HM7OkY1jFw
[3] https://arxiv.org/abs/2410.18693
[4] https://arxiv.org/abs/2406.20094
[5] https://arxiv.org/abs/2407.13690

**Questions:**

* In the first line of Section 4.1, the authors claim that AM-Qwen3-Distilled is "is well-regarded for its high quality and has demonstrated effectiveness in training high quality mathematical reasoning models". Can the authors provide references for this claim?
* Was temperature experimented with during problem generation phase? For thinking models such as Qwen3-8B, a temperature of 0.6 is usually recommended. Is there any reason a temperature of 1.0 was used instead?
* For solution generation - what were the sampling parameters used for Qwen3-8B?
* For solution distillation, what does "reliably" mean? What was the response length of the base model during generation? How was repetition detected?
* Why are 3 solutions distilled per problem in Section 5.2? How are they used (given the final data size in Table 3 is still 192K)

---

> ### Author Response · Authors · 2025-11-17
>
> Thanks for your insightful comments. We’d like to address your concerns as follows.
> > W1: The claim that the thinking vs non-thinking mode switching of AdaptThink can be used as a proxy to judge difficulty is not well substantiated. The authors demonstrate the effectiveness of different subsets of the dataset filtered by AdaptThink as SFT data in Table 1. However, downstream performance cannot be considered as an indicator for the difficulty itself, since it would depend on many factors other than the difficulty of the questions itself, such as but not limited to: diversity of the problems, quality of solution traces, etc. In general, I believe all the factors affecting the usefulness of a problem set as finetuning data are not well understood, thereby introducing confounders. A more direct way of substantiating the claim would be to measure Pass@k of the filtered data subsets under multiple models.
> >
>
> We appreciate your thoughtful critique. We agree that downstream SFT performance can be influenced by multiple factors—problem difficulty, problem diversity, and solution-trace quality. Our goal is to isolate the effect of **difficulty**, and we therefore explicitly control the other variables as follows:
>
> **(1) Solution-trace quality is controlled.**
>
> All solutions in the original dataset $\mathcal{D}$ (AM-Qwen3-Distilled) are distilled **from the same teacher model** (Qwen3-235B-A22B). Thus, solution quality is held constant across $\mathcal{D}\_{Simp}$ and $\mathcal{D}\_{Diff}$.
>
> **(2) Problem diversity is controlled and quantified.**
>
> To quantify the diversity of simple problem set $\mathcal{P}\_{Simp}$ and difficult problem set $\mathcal{P}\_{Diff}$, we embed all problems using Qwen3-Embedding-0.6B. Then we compute *global cosine diversity*, defined as  ${Div}\_{Global} = 1 - \mathbb{E}\left[\cos(x, \bar{x})\right]$, where $\bar{x}$ is the mean embedding vector and all embeddings all L2-normalized.  ${Div}_{Global}$ correlates positively with distributional spread. **The ${Div}_{Global}$ of $\mathcal{P}\_{Simp}$ and $\mathcal{P}\_{Diff}$ are 0.495 and 0.473, respectively, indicating that diversity is adequately controlled.**
>
> We also downsample 10% of $\mathcal{P}\_{Simp}$ and $\mathcal{P}\_{Diff}$ and visualize  their embeddings via t-SNE. From the t-SNE (shown in Appendix Section C of revised manuscript), we can find that $\mathcal{P}\_{Simp}$ and $\mathcal{P}\_{Diff}$ exhibit distinct distributions.
>
> This indicates that the simple subset is not suffering from low diversity, and diversity is not a confounder here.
>
> **We further examine the difficulty characteristics of the data through Pass@k performance and existing difficulty annotations.**
>
> **(1) Direct difficulty measurement via Pass@k.**
>
> Pass@k is commonly regarded as an upper bound on a model’s reasoning capability. Following your recommendation, we further assess difficulty by evaluating Pass@k performance on the problems from $\mathcal{D}\_{Simp}$ and $\mathcal{D}\_{Diff}$.
>
> We randomly sample 2K problems from $\mathcal{D}\_{Simp}$ and $\mathcal{D}\_{Diff}$ and test **three models** with two CoT types: Qwen2.5-Math-7B-Instruct, Qwen3-8B under NoThinking and Thinking mode, and DeepSeek-R1-Distill-Qwen-7B. We set k=3 for all experiments.
>
> As shown in the following table, we can draw the following two conclusions.
>
> - All models exhibit substantially lower pass@3 on the Difficult-2K subset.
> - The gap is especially large for Short-CoT models, which are more sensitive to reasoning difficulty.
>
> | Model | CoT Type | $\mathcal{D}\_{Simp}$-2K | $\mathcal{D}\_{Diff}$-2K |
> | --- | --- | --- | --- |
> | Qwen2.5-Math-7B-Instruct | Short | 88.2 | 65.8 |
> | Qwen3-8B (NoThinking) | Short | 85.8 | 71.5 |
> | Qwen3-8B (Thinking) | Long | 91.1 | 86.0 |
> | DeepSeek-R1-Distill-Qwen-7B | Long | 93.4 | 85.0 |
>
> **(2) Additional validation using external difficulty annotations.**
>
> We further analyze the relationship between AdaptThink’s mode switch and difficulty levels labeled in the test set of MATH [1].
>
> The difficulty levels of MATH range from 1 to 5, which is labeled according to [AoPS](https://artofproblemsolving.com).
>
> From the statistics in following table, we observe a clear monotonic trend that the proportion of “thinking-mode” predictions **increases steadily** with the annotated difficulty level. This indicates that AdaptThink **adaptively selects** its reasoning mode in alignment with established difficulty labels, providing additional support beyond downstream SFT metrics.
>
> | Difficult Level | Thinking Ratio | NoThinking Ratio |
> | --- | --- | --- |
> | 1 | 2.7 | 97.3 |
> | 2 | 4.6 | 95.4 |
> | 3 | 10.9 | 89.1 |
> | 4 | 21.7 | 78.3 |
> | 5 | 41.5 | 58.5 |
>
> These new experiments support our claim that the AdaptThink Thinking/NoThinking switch provides a **meaningful signal of inherent question difficulty**.
>
> We have incorporated these analyses, pass@3 results, and visualizations into Section 3.1.1, Appendix Section B and C of revised manuscript.

---

> ### Author Response · Authors · 2025-11-17
>
> > W2: Solution verification
> >
>
> Thank you for the insightful suggestion. We agree that solution verification is an important consideration. In fact, we preliminarily experimented with a self-consistency–based filter: for each generated problem, we sampled 8 candidate solutions and retained the reasoning trace associated with the majority answer. As shown in the following Table, this self-consistency filtering does not yield measurable performance gains. We hypothesize that this is because the majority vote often collapses to superficial agreement rather than deeper correctness for highly challenging problems.
>
> Given the additional computational overhead and the lack of empirical benefit, we opted for the more efficient one-shot distillation strategy.
>
> |  | AIME'24 (avg@10) | AIME'25 (avg@10) | HMMT-Feb'25 (avg@10) | BRUMO'25 (avg@10) | MATH500 (avg@3) | AVG |
> | --- | --- | --- | --- | --- | --- | --- |
> | ScaleDiff | 73.0±5.0 | 58.7±8.2 | 43.3±4.2 | 66.7±2.7 | 95.2±0.3 | 65.9 |
> | ScaleDiff (Self-Consistency) | 71.0±4.5 | 60.3±7.2 | 39.7±4.1 | 66.0±8.4 | 95.7±1.4 | 65.5 |
>
> > W3: Sampling parameters for the solution distillation
> >
> >
> > W4: Details for the filtering process
> >
> > Q3: For solution generation - what were the sampling parameters used for Qwen3-8B?
> >
> > Q4: What was the response length of the base model during generation? How was repetition detected?
> >
>
> For solution distillation, we follow the official Qwen3 *best practices* for thinking-mode generation (https://huggingface.co/Qwen/Qwen3-8B#best-practices). Specifically, the solution-distillation stage uses a temperature of 1.0, top-p of 0.95, top-k of 20, and output length of 32,768 tokens.
>
> For rule-based solution filtering, we apply purely heuristic filters without using any LLM-as-a-judge. Specifically, we remove (i) overly verbose solutions whose total length exceeds 32,768 tokens or whose outputs do not contain the required `</think>` tag, and (ii) highly redundant solutions exhibiting excessive repetition, defined as any 20-token n-gram occurring more than 20 times within the same solution.
>
> We have added these details to Section 4.1 of the revised manuscript.
>
> > W5: For model based-filtering, using agreement with the teacher model as a criteria for filtering presumes that all the solutions generated by the teacher model are correct, which is not guaranteed or even attempted to be ensured since there is no solution verification involved.
> >
> >
> > Q4: For solution distillation, what does "reliably" mean?
> >
>
> The correct answer to the generated problem is inherently unknown and cannot be determined. Our model-based filtering does not assume that the teacher model’s solutions are correct. Its purpose is to improve data efficiency by discarding problems that the base model already solves *consistently*, regardless of whether the teacher's answer is consistently correct or consistently incorrect. Intuitively, problems on which the base model and the teacher disagree provide stronger training signals.
>
> To further support this interpretation, we conduct a more detailed ablation following the setup in Section 4.3: when removing only the rule-based filtering, the average performance drops to 58.1. As shown in Table 3, removing both rule-based and model-based filtering leads to a further decrease to 55.3. This demonstrates the independent contribution of model-based filtering to overall performance.
>
> We clarify this point explicitly and modify the wording in Section 3.3 and 4.3 of the revised manuscript.

---

> ### Author Response · Authors · 2025-11-17
>
> > W6: I am skeptical of the sole use of Qwen2.5-7B-Math-Instruct as the base models given the recent results these models tend to improve even without data lacking any correct labels as a result of their heavy math oriented pre / post-training. Ideally, the effectiveness of the data should be demonstrated on multiple (atleast 3) base models, preferable across different model families such as Llama and Gemma family of models.
> >
>
> Thank you for the constructive suggestion. To better assess the generality of our ScaleDiff pipeline, despite Qwen2.5, we conduct additional experiments on two more model families—**Llama-3.1** [2] and **DeepSeek-Math** [3]-as our student model. Due to time constraints of rebuttal, we evaluate two settings on a representative subset of our data: (i) training solely on $\mathcal{D}\_{Diff}$ (192K), and (ii) training on $\mathcal{D}\_{\text{Diff}} \cup \mathcal{D}\_{\text{DiffGen}}$ (192K+192K), to measure the effect of data augmentation.
>
> The results summarized in the table below consistently show that incorporating ScaleDiff-generated data yields improvements across these model families. This demonstrates that the effectiveness of ScaleDiff is not tied to the Qwen family or its math-oriented pre/post-training, but transfers robustly to diverse base models.
>
> We have included these results across different model families to Appendix Section D of the revised manuscript.
>
> | Model | Training Set | Training Set Size | AIME'24 (avg@10) | AIME'25 (avg@10) | HMMT-Feb'25 (avg@10) | BRUMO'25 (avg@10) | MATH500 (avg@3) | AVG |
> | --- | --- | --- | --- | --- | --- | --- | --- | --- |
> | Qwen2.5-Math-7B-Instruct | - | - | 11.3±2.7 | 11.3±3.1 | 2.0±1.6 | 18.0±6.0 | 82.7±0.2 | 22.8 |
> | ScaleDiff-Qwen2.5-Math-7B-Instruct | $\mathcal{D}\_{Diff}$ | 192K | 62.3±5.0 | 44.3±7.6 | 36.0±5.7 | 59.0±6.3 | 93.9±1.2 | 56.6 |
> | ScaleDiff-Qwen2.5-Math-7B-Instruct | $\mathcal{D}\_{\text{Diff}} \cup \mathcal{D}\_{\text{DiffGen}}$ | 364K | 65.0±5.2 | 52.0±5.0 | 35.0±3.1 | 62.0±5.0 | 94.0±0.6 | 60.0 |
> | Llama-3.1-8B-Instruct | - | - | 5.3±3.4 | 0.0±0.0 | 0.7±1.3 | 2.7±2.9 | 48.2±1.1 | 9.5 |
> | ScaleDiff-Llama-3.1-8B-Instruct | $\mathcal{D}\_{Diff}$ | 192K | 41.3±6.0 | 28.3±5.6 | 24.7±6.0 | 46.0±9.5 | 86.7±1.1 | 42.6 |
> | ScaleDiff-Llama-3.1-8B-Instruct | $\mathcal{D}\_{\text{Diff}} \cup \mathcal{D}\_{\text{DiffGen}}$ | 364K | 46.3±6.4 | 41.3±6.4 | 28.3±4.5 | 49.7±5.9 | 88.2±1.6 | 49.2 |
> | deepseek-math-7b-instruct | - | - | 0.7±1.3 | 1.0±1.5 | 0.0±0.0 | 1.7±2.2 | 44.7±0.8 | 8.2 |
> | ScaleDiff-deepseek-math-7b-instruct | $\mathcal{D}\_{Diff}$ | 192K | 27.3±6.8 | 25.7±4.7 | 16.0±3.6 | 33.7±4.6 | 83.1±1.1 | 35.3 |
> | ScaleDiff-deepseek-math-7b-instruct | $\mathcal{D}\_{\text{Diff}} \cup \mathcal{D}\_{\text{DiffGen}}$ | 364K | 41.7±4.8 | 27.0±4.8 | 23.7±4.1 | 41.0±8.8 | 87.9±1.1 | 41.4 |
>
> > W7: Data-sized match comparison with other approaches for synthetic data generation
> >
>
> Thank you for the thoughtful suggestion. While these works indeed release their datasets, a direct data-sized matched comparison would be **inherently unfair** **and hard** for two reasons:
>
> **(1) Different seed datasets.** Methods such as ScaleQuest [4], DART-Math [5], and MetaMath [6] are all trained using GSM8K and MATH as seed corpora, whereas our approach relies on a distinct seed distribution. Matching only on dataset size would therefore conflate differences arising from the underlying seed data.
>
> **(2) Different teacher models for distillation.** The compared methods use heterogeneous teacher models—for example, ScaleQuest uses Qwen2-Math-7B-Insruct, DART-Math uses DeepSeekMath-7B-RL, and MetaMath relies on GPT-3.5. Their data quality is thus shaped by model-specific capabilities, making size-based comparisons uninterpretable.
>
> For reasons above, in the main paper we instead compare against strong math-reasoning models that share the **same or closely related base model families**, providing a cleaner and more meaningful evaluation of our method’s effectiveness.
>
> Following your suggestion, we additionally include **MetaMath** [6] as a synthetic generation baseline. We use $\mathcal{D}\_{\text{Diff}}$ as the seed dataset—identical to our DiffGen training—and apply the MetaMath question rephrasing method. As shown in the newly added table, our method outperforms MetaMath under matched settings, further supporting the advantages of ScaleDiff.
>
> | Model | Training Set | Training Set Size | AIME'24 (avg@10) | AIME'25 (avg@10) | HMMT-Feb'25 (avg@10) | BRUMO'25 (avg@10) | MATH500 (avg@3) | AVG |
> | --- | --- | --- | --- | --- | --- | --- | --- | --- |
> | ScaleDiff | $\mathcal{D}\_{Diff}$ | 192K | 62.3±5.0 | 44.3±7.6 | 36.0±5.7 | 59.0±6.3 | 93.9±1.2 | 56.6 |
> | ScaleDiff | $\mathcal{D}\_{\text{Diff}} \cup \mathcal{D}\_{\text{DiffGen}}$ | 364K | 65.0±5.2 | 52.0±5.0 | 35.0±3.1 | 62.0±5.0 | 94.0±0.6 | 60.0 |
> | MetaMath | $\mathcal{D}\_{\text{Diff}} \cup \mathcal{D}\_{\text{MetaMath}}$ | 364K | 62.5±6.5 | 45.0±2.1 | 35.7±4.7 | 59.3±3.6 | 94.3±0.2 | 57.0 |

---

> ### Author Response · Authors · 2025-11-17
>
> > W7: Data-sized match comparison with other approaches for synthetic data generation
> >
>
> Thank you for the thoughtful suggestion. While these works indeed release their datasets, a direct data-sized matched comparison would be **inherently unfair** **and hard** for two reasons:
>
> **(1) Different seed datasets.** Methods such as ScaleQuest [4], DART-Math [5], and MetaMath [6] are all trained using GSM8K and MATH as seed corpora, whereas our approach relies on a distinct seed distribution. Matching only on dataset size would therefore conflate differences arising from the underlying seed data.
>
> **(2) Different teacher models for distillation.** The compared methods use heterogeneous teacher models—for example, ScaleQuest uses Qwen2-Math-7B-Insruct, DART-Math uses DeepSeekMath-7B-RL, and MetaMath relies on GPT-3.5. Their data quality is thus shaped by model-specific capabilities, making size-based comparisons uninterpretable.
>
> For reasons above, in the main paper we instead compare against strong math-reasoning models that share the **same or closely related base model families**, providing a cleaner and more meaningful evaluation of our method’s effectiveness.
>
> Following your suggestion, we additionally include **MetaMath** [6] as a synthetic generation baseline. We use $\mathcal{D}\_{\text{Diff}}$ as the seed dataset—identical to our DiffGen training—and apply the MetaMath question rephrasing method. As shown in the newly added table, our method outperforms MetaMath under matched settings, further supporting the advantages of ScaleDiff.
>
> | Model | Training Set | Training Set Size | AIME'24 (avg@10) | AIME'25 (avg@10) | HMMT-Feb'25 (avg@10) | BRUMO'25 (avg@10) | MATH500 (avg@3) | AVG |
> | --- | --- | --- | --- | --- | --- | --- | --- | --- |
> | ScaleDiff | $\mathcal{D}\_{Diff}$ | 192K | 62.3±5.0 | 44.3±7.6 | 36.0±5.7 | 59.0±6.3 | 93.9±1.2 | 56.6 |
> | ScaleDiff | $\mathcal{D}\_{\text{Diff}} \cup \mathcal{D}\_{\text{DiffGen}}$ | 364K | 65.0±5.2 | 52.0±5.0 | 35.0±3.1 | 62.0±5.0 | 94.0±0.6 | 60.0 |
> | MetaMath | $\mathcal{D}\_{\text{Diff}} \cup \mathcal{D}\_{\text{MetaMath}}$ | 364K | 62.5±6.5 | 45.0±2.1 | 35.7±4.7 | 59.3±3.6 | 94.3±0.2 | 57.0 |
>
> > W8: Comparison with the teacher model performance
> >
>
> Thank you for the valuable comment. Our goal in Section 4.2 is to demonstrate that **ScaleDiff effectively transfers mathematical reasoning ability from a strong teacher model (Qwen3-8B, 68.9 average performance) to a substantially weaker student model (Qwen2.5-7B-Math-Instruct, 22.8 average performance)**. This highlights the effectiveness of our difficulty-hiking data augmentation strategy.
>
> To further address your concern, we additionally conduct distillation experiments starting from the **teacher's own base model, Qwen3-8B-Base**, as the student initialization. The resulting ScaleDiff (Qwen3-8B-Base) model achieves an average performance comparable to our original ScaleDiff (Qwen2.5-7B-Math-Instruct) and slightly below the teacher model. This indicates that the gains are primarily attributable to the **generated difficult queries**, rather than peculiarities of a particular base model.
>
> We also emphasize that the SFT datasets for Qwen3-8B are closed-source, whereas our generated difficult problems will be released to the research community, enabling transparent and reproducible evaluation.
>
> |  | AIME'24 (avg@10) | AIME'25 (avg@10) | HMMT-Feb'25 (avg@10) | BRUMO'25 (avg@10) | MATH500 (avg@3) | AVG |
> | --- | --- | --- | --- | --- | --- | --- |
> | Qwen3-8B | 75.0±4.5 | 64.7±6.4 | 44.0±4.4 | 68.0±2.7 | 96.8±0.3 | 68.9 |
> | ScaleDiff (Qwen2.5-7B-Math-Instruct) | 73.0±5.0 | 58.7±8.2 | 43.3±4.2 | 66.7±2.7 | 95.2±0.3 | 65.9 |
> | ScaleDiff (Qwen3-8B-Base) | 71.7±4.8 | 59.0±8.2 | 41.0±3.7 | 68.0±4.3 | 94.9±0.8 | 65.6 |
>
> > W9: Small differences between means and high standard deviations in Table 3 results
> >
>
> We agree that small absolute differences accompanied by large standard deviations warrant caution in interpreting statistical significance. As highlighted by [7], mathematical-reasoning benchmarks are highly sensitive to variance introduced by random seeds, small test sets, and subtle implementation factors. In particular, they show that Pass@1 estimates can fluctuate by **5–15 percentage points** across seeds, and that on datasets of **only 30 samples**, *“even one question can shift Pass@1 by 2.5–3.3 percentage points,”* making single-seed or low-replication evaluations statistically unstable. The paper therefore emphasizes that *“accurate reporting requires averaging over multiple seeds”* and warns that small datasets make model comparisons inherently unreliable. In light of these insights, our evaluation follows the same rationale proposed by [7].

---

> ### Author Response · Authors · 2025-11-17
>
> > W10: Comparison of the response lengths of $D^{L}\_{Diff}$ and $\hat D^{S}\_{DiffGen}$
> >
>
> Thank you for pointing out this potential confounder. Our conclusion in point (3) is grounded in the preceding analysis in point (2), where we show that **teacher models from the same family (Qwen3) exhibit very similar response-length distributions on difficult problems**.
>
> In other words, once point (2) establishes that Qwen3-family teachers produce length distributions that are highly stable for difficult queries, the comparison between $D^{L}\_{\text{Diff}}$ and $D^{S}\_{\text{DiffGen}}$ becomes less sensitive to the specific choice of teacher model.
>
> We clarify this logical connection explicitly in Section 5.3 of the revised manuscript.
>
> > Other Comments
> >
> > - Citations missing for AIME, MATH, BRUMO, HMMT in the intro.
> > - MATH500 citation is wrong.
>
> Thanks for carefully checking the references. In the revised version, we have corrected the MATH500 citation and added the missing citations for AIME, MATH, BRUMO, and HMMT in the introduction.
>
> > Q1: In the first line of Section 4.1, the authors claim that AM-Qwen3-Distilled is "is well-regarded for its high quality and has demonstrated effectiveness in training high quality mathematical reasoning models". Can the authors provide references for this claim?
> >
>
> Thank you for the question. Our statement is grounded in two sources of evidence:
>
> **(1) Benchmark performance of AM-Thinking-v1-Distilled-math.**
>
> The math subset of the AM-Thinking-v1-Distilled dataset achieves strong results across multiple math reasoning benchmarks, as documented in the **OpenDataArena Leaderboard[8].**
>
> **(2) Effectiveness demonstrated in prior models.**
>
> The **K2-Think [9]** model uses the AM-Thinking-v1-Distilled dataset for SFT and reports strong performance on AIME’24 and AIME’25, further supporting the quality of the underlying data.
>
> Since **AM-Qwen3-Distilled** shares the *same problem set* as AM-Thinking-v1-Distilled (differing only in the teacher model used for distillation—Qwen3-235B-A22B vs. AM-Thinking-v1), and because our teacher model is also from the Qwen3 family (Qwen3-8B), we selected the math subset of AM-Qwen3-Distilled to ensure architectural consistency.
>
> > Q2: Was temperature experimented with during problem generation phase? For thinking models such as Qwen3-8B, a temperature of 0.6 is usually recommended. Is there any reason a temperature of 1.0 was used instead?
> >
>
> Thank you for the question. We would like to clarify that we did **not** use the pretrained Qwen3-8B Thinking mode directly for problem generation. Instead, we train a dedicated difficult-question generator **DiffGen** on Qwen3-8B-Base, using only the filtered difficult problems for training.
>
> In our preliminary experiments, we observed that a **temperature of 1.0** produced substantially more **diverse and challenging** problem formulations compared to a lower temperature such as 0.6. Since the goal of DiffGen is to broaden the difficulty spectrum, this higher temperature leads to noticeably better coverage of difficult problem types.
>
> > Q5: Why are 3 solutions distilled per problem in Section 5.2? How are they used (given the final data size in Table 3 is still 192K)
> >
>
> Thank you for raising this point. We apologize for the missing details in the original text.
>
> The 192K difficult problems in our dataset come with correct solutions generated by Qwen3-235B-A22B. However, for a subset of *extremely* difficult problems, Qwen3-8B is unable to produce a correct solution even after multiple distillation attempts. Simply discarding these problems would introduce bias toward easier instances, whereas retaining these incorrect solutions raises concerns about reasoning quality.
>
> To balance these considerations, we distill 3 solutions per problem. If at least one of the three is correct, we retain one correct solution; otherwise, we randomly select one of the three distilled attempts. In all cases, **only a single solution per problem** is included in the final dataset, which explains why the final size remains 192K.
>
> We have clarified this point in Section 5.2 of the revised manuscript.

---

> ### Author Response · Authors · 2025-11-17
>
> **References**
>
> [1] Hendrycks, Dan, et al. "Measuring Mathematical Problem Solving With the MATH Dataset." *Thirty-fifth Conference on Neural Information Processing Systems Datasets and Benchmarks Track (Round 2)*.
>
> [2] https://ai.meta.com/blog/meta-llama-3-1/
>
> [3] Shao, Zhihong, et al. "Deepseekmath: Pushing the limits of mathematical reasoning in open language models." *arXiv preprint arXiv:2402.03300* (2024).
>
> [4] Ding, Yuyang, et al. "Unleashing LLM Reasoning Capability via Scalable Question Synthesis from Scratch." *Proceedings of the 63rd Annual Meeting of the Association for Computational Linguistics (Volume 1: Long Papers)*. 2025.
>
> [5] Tong, Yuxuan, et al. "Dart-math: Difficulty-aware rejection tuning for mathematical problem-solving." *Advances in Neural Information Processing Systems* 37 (2024): 7821-7846.
>
> [6] Yu, Longhui, et al. "MetaMath: Bootstrap Your Own Mathematical Questions for Large Language Models." *The Twelfth International Conference on Learning Representations*.
>
> [7] Hochlehnert, Andreas, et al. A Sober Look at Progress in Language Model Reasoning: Pitfalls and Paths to Reproducibility. *Second Conference on Language Modeling.* 2025
>
> [8] https://opendataarena.github.io/leaderboard.html
>
> [9] Cheng, Zhoujun, et al. "K2-think: A parameter-efficient reasoning system." *arXiv preprint arXiv:2509.07604* (2025).

---

> ### Author Response · Authors · 2025-11-26
>
> Dear Reviewer s7ya,
>
> Thank you again for your time and constructive feedback.
>
> We have posted a detailed response to your review comments a few days ago, addressing your concerns one by one.
> We would firmly appreciate it if you could let us know whether our response has sufficiently addressed your concerns or if there are any remaining questions we can clarify before the discussion period ends.
>
> Best regards, The Authors

---

### Note · Authors · 2026-01-04

**Comment:**

We would like to sincerely thank the area chair and all reviewers for their time and the constructive feedback provided during the review process. We have decided to withdraw our submission to further improve the work based on your valuable suggestions. We appreciate your efforts in helping us enhance the quality of our research.

**Withdrawal Confirmation:**

I have read and agree with the venue's withdrawal policy on behalf of myself and my co-authors.